# Revisiting Regularized Policy Optimization for Stable and Efficient Reinforcement Learning in Two-Player Games

**Kazuki Ota** [1 2]  **Takayuki Osa** [2]  **Motoki Omura** [1]  **Tatsuya Harada** [1 2]

## Abstract

Two-player games such as board games have long been used as traditional benchmarks for reinforcement learning. This work revisits a policy optimization method with reverse Kullback-Leibler regularization and entropy regularization and analyzes this combination in two-player zero-sum settings from theoretical and empirical perspectives. From a theoretical perspective, we investigate the stability of the policy update rule in two theoretical settings: game-theoretic normal-form games and finite-length games. We provide novel convergence guarantees and verify our theoretical results through numerical experiments on synthetic games. From an empirical perspective, we derive a practical model-free reinforcement learning algorithm based on the regularized policy optimization. We validate the training efficiency of our algorithm through comprehensive experiments on five board games: Animal Shogi, Gardner Chess, Go, Hex, and Othello. Experimental results show that our agent learns more efficiently than existing methods across environments.

## 1. Introduction

In the domain of two-player zero-sum games, search-based reinforcement learning (RL) methods, exemplified by AlphaZero (Silver et al., 2018), have established themselves as the state-of-the-art. While these approaches have achieved superhuman performance by combining deep neural networks with look-ahead search including Monte-Carlo Tree Search (MCTS), their success comes at a substantial computational cost. Indeed, it has been reported that the training process of AlphaZero requires more than 10 GPU-years to converge (Silver et al., 2018; Tian et al., 2019). This massive

demand for computational resources often hinders reproducibility and practical application in resource-constrained environments (Zhao et al., 2022).

To address this issue, recent studies have proposed modified search-based algorithms that reduce the depth and rollout count of the tree search during training (Hessel et al., 2021; Danihelka et al., 2022). While these methods mitigate the computational burden, they still fundamentally rely on look-ahead search to construct improved policy targets. In contrast, model-free RL methods, which are dominant in robotics and continuous control due to their simplicity and computational efficiency (Kroemer et al., 2021; Tang et al., 2025), have not been fully explored in board games due to perceived instability in multi-agent settings. This raises a fundamental question: *Can we design a pure model-free RL algorithm that achieves stable and competitive learning with significantly fewer training resources than search-based counterparts?*

We answer this question affirmatively by revisiting regularized policy optimization and analyzing it from both theoretical and empirical perspectives. Inspired by the insight that AlphaZero implicitly solves a policy optimization problem with KL regularization (Grill et al., 2020), we identify that the instability of self-play RL in two-player games can be effectively tamed by a specific combination of regularization techniques: reverse Kullback-Leibler (KL) regularization for gradual policy updates and entropy regularization for sustained exploration. To theoretically validate the stability of this approach, we derive novel convergence guarantees for the resulting update rule in two theoretical settings: normal-form games and finite-length games. We also verify these theoretical results by numerical experiments.

Motivated by the theoretical insights, we investigate a practical model-free RL algorithm based on policy optimization with reverse-KL regularization and entropy regularization, which we refer to as KLENT. We validate the efficiency of our agent through comprehensive experiments on five board games: Animal Shogi, Gardner Chess, 9x9 Go, Hex, and Othello. Without model-based search during training, our agent achieved up to 4x higher training efficiency than existing approaches. Our extensive ablation study verify the importance of each techniques.

[1]The University of Tokyo, Japan [2]RIKEN Center for Advanced Intelligence Project, Japan. Correspondence to: Kazuki Ota <ota@mi.t.u-tokyo.ac.jp>.

*Proceedings of the $43^{rd}$ International Conference on Machine Learning*, Seoul, South Korea. PMLR 306, 2026. Copyright 2026 by the author(s).

The main contributions are summarized as follows:

1. We revisit a regularized policy optimization problem with reverse-KL regularization and entropy regularization and derive policy update rule and practical RL algorithm based on it.

2. From theoretical perspective, we provide novel convergence guarantees on two settings: normal-form games and finite-length games and verify them with numerical experiments.

3. From empirical perspective, we demonstrate that our agent achieves more efficient learning than existing methods through comprehensive experiments on board games.

We would like to clarify that our main contributions do not lie in proposing entirely new algorithmic ingredients. Instead, our contributions are new theoretical and empirical characterizations of this specific regularized policy optimization in two-player games.

Although we focus on board games in this study, our assumption is merely an MDP with a finite action space. This class of problems covers several real-world applications such as discrete optimization, algorithmic discovery, and mathematical proving (Fawzi et al., 2022; Mankowitz et al., 2023; Hubert et al., 2025), where search-based methods such as AlphaZero are widely used. Our efficient algorithm may also achieve efficient learning in these practical domains, accelerating research on such real-world problems.

## 2. Problem Setting

In this study, we formulate two-player zero-sum games such as board games as reinforcement learning problems. Reinforcement learning (RL) (Sutton et al., 1998) is a framework in which an agent learns a policy $\pi$ through interactions with an environment to maximize an expected return. This framework can be formalized as a Markov Decision Process (MDP) (Bellman, 1957), consisting of a state space $\mathcal{S}$, an action space $\mathcal{A}$, a transition probability function $P(s'|s, a)$, a reward function $r(s, a)$, and a discount factor $\gamma \in [0, 1]$. At each time step $t$, the agent selects an action $A_t \in \mathcal{A}$ based on its policy $\pi(A_t|S_t)$ and the current state $S_t \in \mathcal{S}$. In response, the environment transitions to the next state $S_{t+1} \in \mathcal{S}$ according to the transition probability $P(S_{t+1}|S_t, A_t)$ and provides a reward $R_t = r(S_t, A_t)$. The objective of the agent is to maximize the expected return $\mathbb{E}_{(S_t, A_t, R_t) \sim (P, \pi)} \left[ \sum_{t=0}^{T} \gamma^t R_t \right]$. Here, $T$ represents the terminal timestep of an episode. The state-value function $V^\pi(s) = \mathbb{E}_{(P,\pi)}[\sum_{t=0}^{T} \gamma^t R_t | S_0 = s]$ and the action-value function $Q^\pi(s, a) = \mathbb{E}_{(P,\pi)}[\sum_{t=0}^{T} \gamma^t R_t | S_0 = s, A_0 = a]$ can be used to evaluate and improve the policy $\pi$.

Board games are highly complex decision-making tasks as even human experts may spend minutes to hours deliberating on a single action. Due to this complexity, they have served as a canonical benchmark for artificial intelligence for several decades (Samuel, 1959; Tesauro et al., 1995; Campbell et al., 2002; Silver et al., 2016; Yannakakis & Togelius, 2018). Two-player zero-sum games, including board games, can be formulated as an MDP. The reward is assigned as $R_T = +1$ for a win, $R_T = -1$ for a loss, and $R_T = 0$ for a draw, while rewards at all other timesteps $t \in \{0, 1, \ldots, T - 1\}$ are zero. By setting the discount factor $\gamma$ to 1, we ensure that the final outcome of the game is directly reflected in the expected return.

## 3. Related Work

### 3.1. Regularized Policy Optimization

Several RL algorithms have been proposed based on the paradigm of regularized policy optimization, which can generally be formulated as follows:

$$\underset{\pi'}{\text{maximize}} \ \mathbb{E}_{A \sim \pi'(\cdot|s)}[Q^\pi(s, A)] - \mathcal{R}(\pi'). \quad (1)$$

Here, $\pi'$ is the optimized policy, $\pi$ is the prior policy, and $\mathcal{R}(\pi')$ is the regularization term. For example, if we define the regularization term as $\mathcal{R}(\pi') = -\alpha H(\pi')$, where $H(\pi')$ is the entropy of the optimized policy $\pi'$, the optimal solution corresponds to a softmax policy $\pi(a|s) \propto \exp(Q^\pi(s, a)/\alpha)$. This policy has long been adopted in prior studies, including classical approaches such as REINFORCE (Williams, 1992) and SARSA (Rummery & Niranjan, 1994; Van Seijen et al., 2009). Soft Q-Learning (Haarnoja et al., 2017) and SAC (Haarnoja et al., 2018) are methods that treat the entropy term as additional rewards.

Alternatively, if we define the regularization term as the difference between prior policy $\pi$ and optimized policy $\pi'$, we can make the policy updates gradual. For example, TRPO (Schulman et al., 2015) and one variant of PPO (Schulman et al., 2017) use the forward KL divergence $\mathcal{R}(\pi') = \beta D_{\mathrm{KL}}(\pi \| \pi')$ and MPO (Abdolmaleki et al., 2018) use the reverse KL divergence $\mathcal{R}(\pi') = \beta D_{\mathrm{KL}}(\pi' \| \pi)$. Entropy regularization can also be combined to enhance exploration for these methods. Interestingly, Grill et al. (2020) have pointed out that AlphaZero (Silver et al., 2018) is also approximately solving a policy optimization problem with KL regularization.

For the methods that leverage both reverse KL regularization and entropy regularization, Vieillard et al. (2020a), Sokota et al. (2022), and Zhan et al. (2023) provided the theoretical properties of this combination. Following these lines of research, we provide convergence guarantee on this combination of regularizations and validate it with numerical experiments.

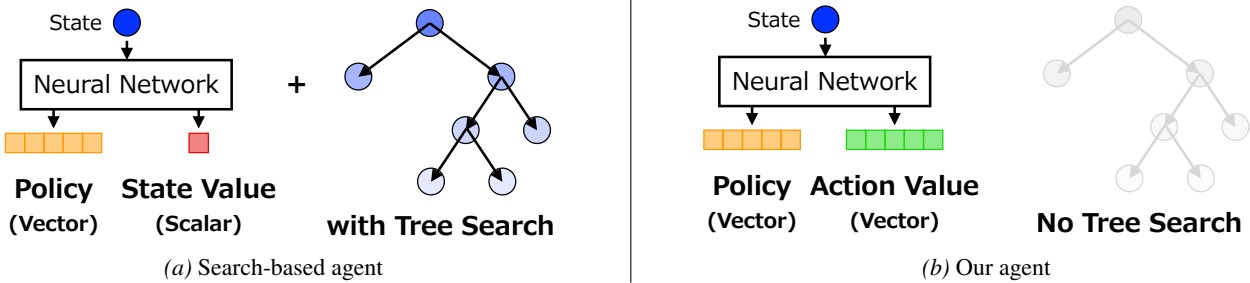

*Figure 1.* Conceptual comparison between search-based approaches and our agent KLENT. *(a)* Search-based methods such as AlphaZero model the policy and the state-value function $V(s)$, and use tree search to estimate the action-value function $Q(s, a)$. *(b)* KLENT, by contrast, directly models both the policy and the action-value function using neural networks, eliminating the need for search.

### 3.2. Self-Play RL in Two-Player Games

Search-based approaches have demonstrated strong performance in two-player games such as board games. One of the most well-known algorithms is AlphaGo (Silver et al., 2016). It combined supervised pre-training with human expert game records and fine-tuning by RL with MCTS, defeating a human world champion in the game of Go. AlphaGo Zero (Silver et al., 2017) eliminated the need for supervised pre-training, and AlphaZero (Silver et al., 2018) extended it to general perfect-information finite-action games. Its generality has enabled applications in other fields, including mathematical and algorithmic discovery (Fawzi et al., 2022; Mankowitz et al., 2023). Subsequent studies of AlphaZero (Schrittwieser et al., 2020; Hubert et al., 2021; Ozair et al., 2021; Schrittwieser et al., 2021) have extended its applicability to a wider range of RL settings, such as continuous action spaces and partial observations.

While these search-based approaches are powerful, their computational demand has been noted as a limitation (Zhao et al., 2022). To address this issue, Hessel et al. (2021) and Danihelka et al. (2022) proposed methods that reduce tree depth and rollout count, respectively. Our work shares the goal of achieving efficient learning with these prior studies, but takes a more drastic approach. While previous methods reduce the amount of look-ahead search during training, we aim to completely eliminate it. In this sense, our method can be regarded as the zero-search limit of this line of research. Conceptual comparison between search-based methods and KLENT is illustrated in Figure 1.

Another line of research aims to enhance game-playing agents by incorporating game-specific knowledge. This approach has been adopted in both perfect-information games (Romstad et al., 2016; Delorme, 2017; Wu et al., 2020) and imperfect-information games (Moravčík et al., 2017; Li et al., 2020; Perolat et al., 2022; Bakhtin et al., 2023), leading to strong performance. However, our aim is to design a game-agnostic pure RL algorithm, which distinguishes our work from these prior studies.

## 4. KL and Entropy Regularized Policy Optimization

In this study, we investigate KL and Entropy Regularized Policy Optimization (KLENT). In Section 4.1, we describe our policy update rule, detailing our policy optimization problem and the solution to it. In Section 4.2, we explain value function learning methodology, utilizing $\lambda$-returns to stabilize the learning process. Section 4.3 presents the practical RL algorithm with a pseudo code.

### 4.1. Policy Update Rule

In regularized policy optimization, it is well established that reverse KL regularization yields gradual updates and entropy regularization maintains exploration. In self-play, optimizing a policy against constantly changing opponents is a non-stationary problem requiring gradual updates to prevent abrupt policy changes. Furthermore, addressing the train-test distribution shift from unseen test-time opponents requires moderate exploration to prevent over-fitting to the policy of the agent itself. Reverse KL and entropy regularizations address non-stationarity and distribution shift, respectively.

Leveraging these two regularizers, we consider the following regularized policy optimization problem.

$$\underset{\pi'}{\text{maximize}} \ \mathbb{E}_{A \sim \pi'(\cdot|s)}[Q^\pi(s, A)]$$
$$- \beta D_{\text{KL}}(\pi'(\cdot|s)\|\pi(\cdot|s)) + \alpha H(\pi'(\cdot|s)). \quad (2)$$

Here, $D_{\text{KL}}(\pi'\|\pi)$ is the reverse KL divergence between the new policy $\pi'$ and the current policy $\pi$, and $H(\pi')$ is the entropy of $\pi'$. The coefficients $\alpha$ and $\beta$ are the non-negative scalar hyperparameters which control the strength of the regularization terms. Leveraging the fact that the action space $\mathcal{A}$ of board games is finite, the optimal solution $\pi'$ can be analytically derived in the following closed-form expression:

$$\pi'(a|s) = \frac{1}{Z(s)} \exp\left(\frac{Q^\pi(s, a) + \beta \log \pi(a|s)}{\alpha + \beta}\right), \quad (3)$$

where $Z(s) = \sum_{a \in \mathcal{A}} \exp\left(\frac{Q^\pi(s,a) + \beta \log \pi(a|s)}{\alpha + \beta}\right)$ is a normalization term to ensure that $\pi'(\cdot|s)$ is a probability distribution. Appendix A provides the detailed derivation of this optimal solution. In KLENT, this analytically obtained policy $\pi'$ is used for action selection during the training.

We model the policy as $\pi_\theta(a|s)$ with a neural network. When updating the parameter $\theta$, the analytically obtained optimal policy $\pi'(\cdot|s)$ is used as the learning target, and fitting of $\theta$ is conducted to minimize the cross-entropy $-\sum_{a \in \mathcal{A}} \pi'(a|s) \log \pi_\theta(a|s)$.

## 4.2. Learning Action-Value Function

In self-play methods for two-player games such as board games, a common approach is to model the state-value function $V^\pi(s)$. One then runs MCTS and backs up state values along future trajectories to estimate action values at the current state. These action-value estimates are used to update the policy at the current state (Silver et al., 2018; Grill et al., 2020; Danihelka et al., 2022). In contrast, KLENT directly parameterize the action-value function $Q^\pi(s, a)$ with a neural network and use it to compute Equation 3, as mentioned in Figure 1. This enables model-free learning without relying on MCTS.

In prior work (Silver et al., 2018; Grill et al., 2020; Danihelka et al., 2022), it is common to use the Monte Carlo return as the learning target for the value function. However, in other areas of reinforcement learning, $\lambda$-returns are also used since they can reduce variance (Sutton, 1988). We conducted preliminary experiments on 9x9 Go using a pretrained checkpoint (Koyamada et al., 2023), with results illustrated in Figure 2. We have confirmed that even in a board game environment, an intermediate $\lambda \in (0, 1)$ minimizes the sum of squared bias and variance better than $\lambda = 1$, which corresponds to the Monte Carlo return. Motivated by this observation, we use the $\lambda$-return $G^\lambda$ as the learning target for the value function in this work.

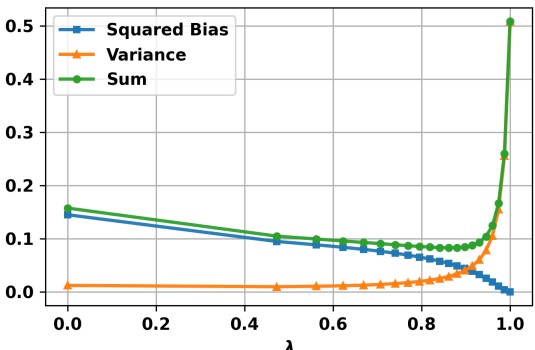

*Figure 2.* Results of preliminary experiments on bias-variance tradeoff in 9x9 Go. Larger $\lambda$ reduces squared bias but increases variance, and an intermediate $\lambda$ minimizes their sum.

## 4.3. Practical Algorithm

The practical algorithm of our agent KLENT is illustrated in Algorithm 1. Starting from randomly initialized networks, KLENT updates the policy $\pi_\theta$ and the action-value function $Q_\theta$ alternating a self-play phase for data collection and a fitting phase for network updates.

In the self-play phase, the goal is to populate the on-policy sample buffer $\mathcal{D}$. During the episode, the actions are sampled from the policy $\pi'$ using the current networks $\pi_\theta$ and $Q_\theta$. After the episode terminates, the $\lambda$-return $G_t^\lambda$ is computed for all timesteps $t$. Samples are collected by repeatedly running episodes until the number of samples in the buffer reaches a predefined capacity.

In the fitting phase, the data accumulated in the buffer $\mathcal{D}$ is used to update the network parameter $\theta$. The loss function $L(\theta)$ is defined as follows:

$$L(\theta) = \mathbb{E}_{\mathcal{D}}\left[ -\sum_{a \in \mathcal{A}} \pi'(a|S) \log \pi_\theta(a|S) + (Q_\theta(S, A) - G^\lambda)^2 \right]. \tag{4}$$

Here, $\mathbb{E}_{\mathcal{D}}[\cdot]$ indicates that $(S, A, (\pi'(a|S))_{a \in \mathcal{A}}, G^\lambda)$ are sampled from the buffer $\mathcal{D}$. This loss function is designed to simultaneously optimize the policy and action-value networks, with the analytically obtained policy $\pi'(\cdot|S)$ and $\lambda$-return $G^\lambda$ serving as targets for learning. By iterating these self-play and fitting phases, the policy $\pi_\theta$ and the action-value function $Q_\theta$ are progressively refined and eventually become strong.

---

**Algorithm 1** KLENT Algorithm

---

1: Initialize the policy network $\pi_\theta(a|s)$.
2: Initialize the action-value network $Q_\theta(s, a)$.
3: **repeat**
4:     $\mathcal{D} \leftarrow \{\}$
5:     **repeat**
6:         Initialize the state $S_0$.
7:         **for** $t = 0, \dots, T$ **do**
8:             $\pi'(a|S_t) \propto \exp\left(\frac{Q_\theta(S_t, a) + \beta \log \pi_\theta(a|S_t)}{\alpha + \beta}\right)$
9:             $\hat{v}_t \leftarrow \mathbb{E}_{A \sim \pi'(\cdot|S_t)}\left[Q_\theta(S_t, A)\right]$
10:        Sample $A_t \sim \pi'(\cdot|S_t)$.
11:        Execute $A_t$ and observe $(S_{t+1}, R_t)$.
12:         **end for**
13:        Compute $\lambda$-returns $\{G_t^\lambda\}_{t=0}^T$.
14:        $\mathcal{D} \leftarrow \mathcal{D} \cup \left\{(S_t, A_t, (\pi'(a|S_t))_{a \in \mathcal{A}}, G_t^\lambda)\right\}_{t=0}^T$
15:     **until** $\mathcal{D}$ reaches a predefined capacity.
16:     Update $\theta$ by minimizing $L(\theta)$ in (4).
17: **until** convergence.

---

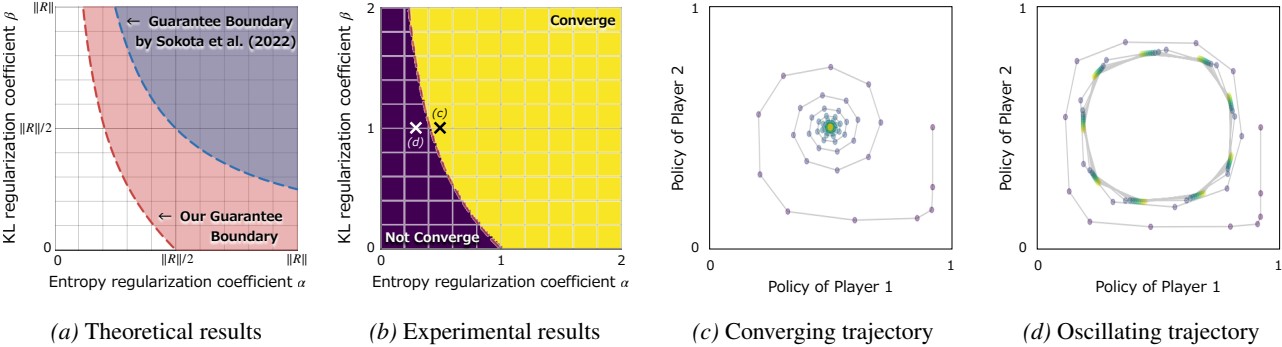

*(a)* Theoretical results     *(b)* Experimental results     *(c)* Converging trajectory     *(d)* Oscillating trajectory

*Figure 3.* Theoretical and experimental results on the normal-form game. *(a)* Theorem 5.1 guarantees convergence in red area. *(b)* The yellow and purple areas indicate convergence and non-convergence in numerical experiments respectively. *(c)* With $(\alpha, \beta) = (0.5, 1)$, which satisfies condition (5), the policies converge. *(d)* With $(\alpha, \beta) = (0.3, 1)$, which violates condition (5), the policies oscillate.

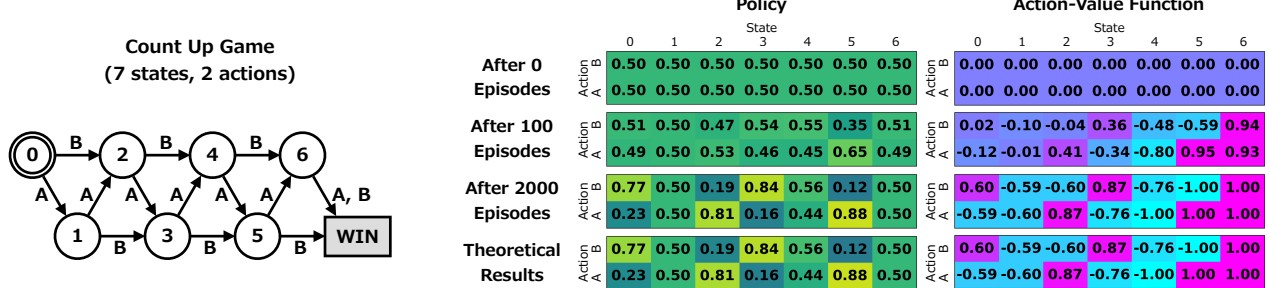

*Figure 4.* Experimental results on the finite-length game. Left: The transition diagram of the Count Up Game with 7 states and 2 actions. Right: The evolution of the policy and action-value function and theoretically expected convergence limit.

## 5. Theoretical Analysis

In this section, we investigate theoretical aspects of KLENT, especially its convergence properties. Section 5.1 provides a theoretical analysis of normal-form games, a standard setting in game theory. Section 5.2 presents an analysis of finite-length games, including sequential turn-based board games. In both sections, we focus specifically on two-player zero-sum games.

### 5.1. Normal-Form Games

In this section, we investigate the convergence property of our policy update rule in normal-form games. This setting is standard in game-theoretic analysis and aligns with the theoretical setting used in prior work (Sokota et al., 2022). We consider a two-player zero-sum game defined by a payoff matrix $R$, where both players update their policies according to Equation 3. Under this setting, the following theorem holds regarding the local linear convergence to the unique fixed point.

**Theorem 5.1.** *The policy update rule in Equation 3 is locally linearly convergent to the unique fixed point if the following condition is satisfied:*

$$\alpha(\alpha + 2\beta) > \|R\|_2^2/4. \tag{5}$$

This condition is illustrated in Figure 3a. For comparison, the convergence condition derived in Sokota et al. (2022) corresponds to $\alpha\beta > \|R\|_2^2$. As shown in the figure, our result covers a broader range of regularization coefficients $(\alpha, \beta)$ than their result. The detailed statement and proof are provided in Appendix C.1. Our proof strategy involves analyzing the spectral radius of the Jacobian of the update operator around the fixed point. We prove that the operator norm is less than 1 under the condition above.

To verify this theoretical result, we conducted numerical experiments on the Matching Pennies game (Gibbons, 1992), defined by $R = \left(\begin{smallmatrix} 1 & -1 \\ -1 & 1 \end{smallmatrix}\right)$, which has a spectral norm of $\|R\|_2 = 2$. We tried various combinations of $\alpha$ and $\beta$ to check convergence and the results are shown in Figure 3b. By comparing the theoretical boundary in Figure 3a and the experimental results in Figure 3b, we observe that the boundary between convergence and divergence in the experiments matches our theoretical condition. It can also be observed that the case $(\alpha, \beta) = (0, 0)$ results in non-convergence, highlighting the necessity of regularization for stable learning. Figures 3c and 3d further illustrate typical policy trajectories, showing stable convergence inside the boundary and limit cycles outside it.

*Table 1.* Five board game environments used for the experiments. Branching factor is the average number of legal actions over states encountered in random rollouts generated by baseline agents in Pgx (Koyamada et al., 2023).

| Game Name | Animal Shogi | Gardner Chess | 9x9 Go | Hex | Othello |
|---|---|---|---|---|---|
| **Initial State** | | | | | |
| **Observation Shape** | (4, 3, 194) | (5, 5, 115) | (9, 9, 17) | (11, 11, 4) | (8, 8, 2) |
| **Action Space Size** | 132 | 1225 | 82 | 122 | 65 |
| **Branching Factor** | 7.5 | 9.5 | 42.3 | 90.6 | 8.0 |

## 5.2. Finite-Length Games

In this section, we discuss the convergence property of the overall KLENT algorithm in sequential two-player zero-sum games. Specifically, we assume that the game length is finite. That is, there exists a positive integer $T_{\max}$ such that the terminal timestep $T$ always satisfies $T \leq T_{\max}$[1]. This assumption can be justified for board games as games like Hex and Othello naturally terminate when the board fills up and others like Chess are typically truncated by rules to ensure finiteness.

Under this assumption, we prove that the policy of the KLENT agent converges to the entropy-regularized optimal policy, which satisfies the following equation:

$$\pi(a|s) = \frac{1}{Z(s)} \exp(Q^\pi(s, a)/\alpha). \quad (6)$$

As $\alpha \to 0$, this regularized equilibrium approaches a Nash equilibrium of the original game (McKelvey & Palfrey, 1995). The detailed statement and proof are provided in Appendix C.2. We prove this convergence via backward induction from terminal states where fixed terminal values propagate stability back to the root.

We empirically verified these theoretical results using a Count Up Game as a synthetic testbed. As illustrated in the left of Figure 4, this game consists of 7 states with 2 actions each, where the player who reaches the WIN state wins the game. We ran KLENT on this environment and presented the evolution of learned policy and action-value function in the right of Figure 4. After 2000 episodes, the learned policy and action-value function align with the theoretical results, confirming the consistency of our analysis. Furthermore, the snapshot at 100 episodes reveals that the policy and values near the terminal states are learned earlier than those near the start, which is consistent with the backward propagation mechanism described in our proof.

---

[1]This assumption implies that the reachable state-transition graph is acyclic, as the presence of a reachable directed cycle would allow trajectories of unbounded length.

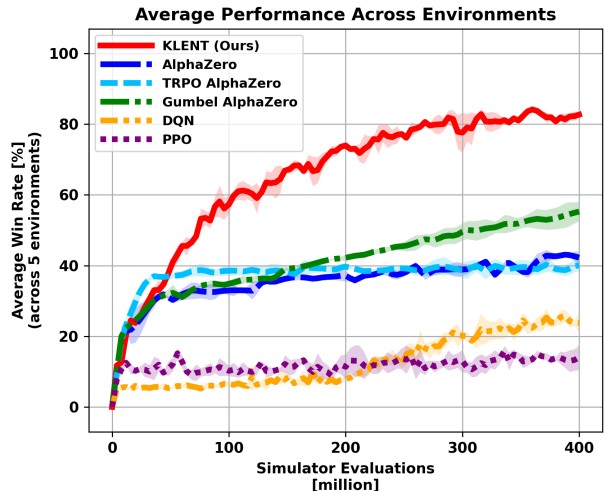

*Figure 5.* Empirical results on learning efficiency. Average performance across five board games. Our agent (KLENT) achieves up to 4x higher training efficiency than Gumbel AlphaZero, the strongest competing baseline in our experiments.

## 6. Experiments

In this section, we present our experimental results on board games. Specifically, Section 6.1 provides the results of performance comparison on five board games, demonstrating the efficiency of KLENT compared to existing methods. Subsequently, we present the results of our ablation study in Section 6.2, demonstrating the importance of the key techniques in KLENT, namely KL regularization, entropy regularization, and $\lambda$-returns. Lastly, we provide the experimental results on large-scale 19x19 Go in Section 6.3.

### 6.1. Performance Comparison

**Setup.** We employed five medium-scale board games listed in Table 1 to compare the performance and the learning efficiency of KLENT and existing approaches. We measured the win rates of each agent against anchored opponents with pretrained checkpoints from Koyamada et al. (2023). For the horizontal axis, we employed the number

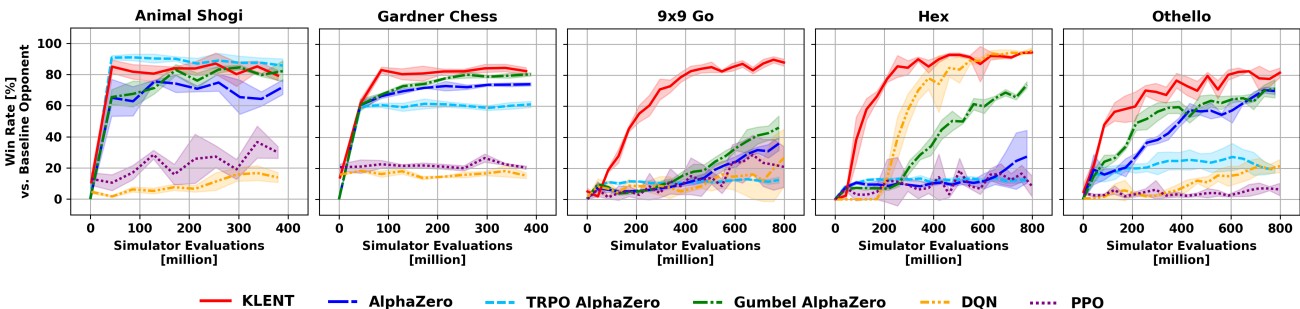

*Figure 6.* Performance comparison between our agent KLENT and existing methods on five board games. KLENT achieves competitive or higher efficiency compared to existing methods.

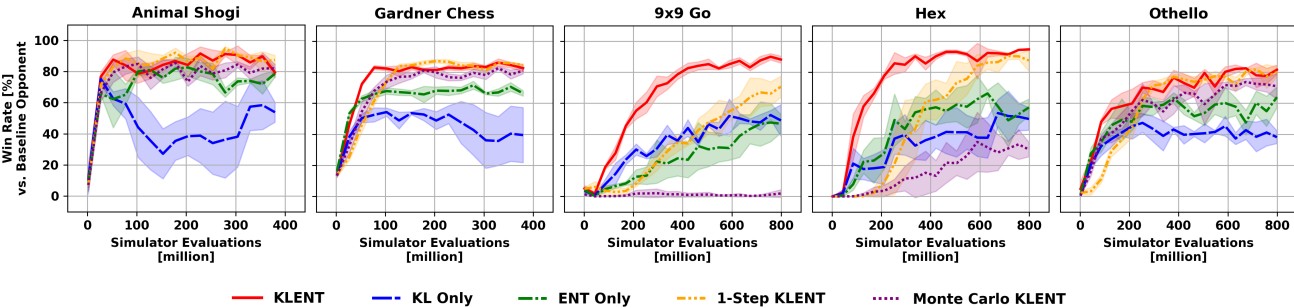

*Figure 7.* The results of the ablation study. They highlight the importance of all three techniques for consistently achieving high efficiency across the environments.

of simulator evaluations which serves as an indicator of the computational demand of training processes and has been adopted in the literature, particularly when training efficiency is of the primary interest (Wu et al., 2020). In the evaluation, we have used a reactive policy for plotting the learning curve in order to unify the test-time computational resources for all methods. As baselines for performance comparison, we used AlphaZero (Silver et al., 2018), TRPO AlphaZero (Grill et al., 2020), and Gumbel AlphaZero (Danihelka et al., 2022) as search-based approaches, and DQN (Mnih et al., 2015) and PPO (Schulman et al., 2017) as model-free approaches. The network architecture was unified across all experiments, specifically utilizing a ResNet (He et al., 2016) with 6 residual blocks. The hyperparameters of KLENT were unified across all five environments and set to $(\alpha, \beta, \lambda) = (0.03, 0.1, e^{-1/8})$. Further details of the experimental setup and implementation are provided in Appendix E and Appendix F, respectively.

**Results.** The average performances across the five environments are presented in Figure 5. The results show that our agent KLENT achieves the most efficient learning on average. In particular, the results indicate several-fold efficiency gains. For example, Gumbel AlphaZero required 300 million simulator evaluations to reach an average win rate of 50%, whereas KLENT required only 75 million, representing a fourfold efficiency gain.

The detailed performances in each environment are also presented in Figure 6. In Animal Shogi and Gardner Chess, where search-based approaches demonstrate high performance with moderate number of simulator evaluations, KLENT achieves competitive efficiency. In 9x9 Go, Hex, and Othello, where search-based approaches require substantial training resources, KLENT demonstrates significantly higher efficiency. This pattern is reflected in the branching-factor statistics in Table 1, where the branching factor is measured as the mean number of legal actions. In Animal Shogi and Gardner Chess, where the branching factors are 7.5 and 9.5, KLENT and search-based methods are competitive. In contrast, in 9x9 Go and Hex, where the factors are 42.3 and 90.6, KLENT shows a clearer advantage. This is consistent with the intuition that larger branching factors increase MCTS simulator-evaluation budgets, whereas KLENT avoids look-ahead search.

We attribute the efficiency of KLENT to two design choices. First, KLENT updates the policy analytically via regularization, rather than using MCTS outputs, which reduces simulator evaluations and neural network inferences per decision. Second, KLENT uses $\lambda$-returns as value targets instead of the Monte Carlo returns commonly used in search-based methods, improving target stability in value learning; our ablation results support this effect. Together, analytical policy updates and stable value targets appear to be the main drivers of KLENT's high learning efficiency.

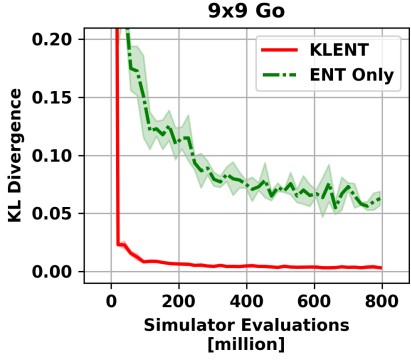
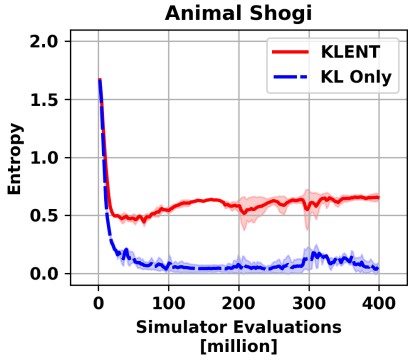
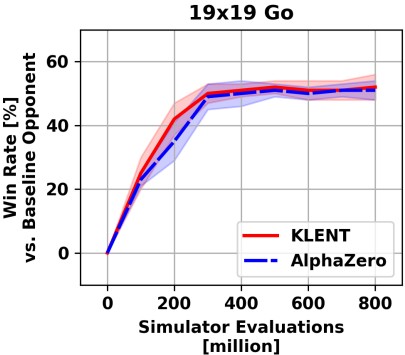

*Figure 8.* The evolution of KL Divergence $D_{\mathrm{KL}}(\pi'\|\pi)$ in 9x9 Go. In KLENT, reverse-KL regularization keeps it relatively low, resulting in gradual policy updates.

*Figure 9.* The evolution of policy entropy $H(\pi')$ in Animal Shogi. While KLENT maintains the entropy, it becomes nearly zero in KL Only.

*Figure 10.* The results of experiments in 19x19 Go. Even in the large-scale environment, KLENT achieves competitive learning efficiency compared to AlphaZero.

We also evaluated AlphaZero, Gumbel AlphaZero and KLENT in matches with test-time MCTS. For each method, we used the checkpoint trained after 800 million simulator evaluations. For KLENT, we adopted the Gumbel AlphaZero MCTS at test time. All agents including baseline opponent used MCTS with 800 rollouts in matches. The results in Table 2 show that KLENT attains high win rates in average even when equipped with test-time MCTS.

*Table 2.* Win rates of each agent against the baseline agent in matches using test-time MCTS. Each agent is trained with 800 million simulator evaluations. Average and standard error are shown. "AZ" stands for AlphaZero.

|               | AZ       | Gumbel AZ | KLENT    |
|---------------|----------|-----------|----------|
| Animal Shogi  | 31±2%    | **67±5%** | 63±4%    |
| Gardner Chess | 64±3%    | 70±1%     | **81±1%**|
| 9x9 Go        | 7±2%     | 37±2%     | **89±1%**|
| Hex           | 8±5%     | 47±5%     | **98±1%**|
| Othello       | 51±2%    | 47±3%     | **55±6%**|
| Average       | 32.2%    | 53.6%     | **77.2%**|

### 6.2. Ablation Study

**Setup.** We conducted an ablation study to validate the importance of the three key techniques in KLENT: KL regularization, entropy regularization, and the use of $\lambda$-returns. We compared KLENT with the following four variants. **KL Only**: Entropy regularization is removed by setting $\alpha = 0$. **ENT Only**: KL regularization is removed by setting $\beta = 0$. **1-Step KLENT**: $\lambda$-returns are replaced with 1-step backups by setting $\lambda = 0$. **Monte Carlo KLENT**: $\lambda$-returns are replaced with Monte Carlo returns by setting $\lambda = 1$.

**Results.** The results of our ablation study are shown in Figure 7. The results demonstrate the importance of all three techniques for consistently achieving high efficiency in the

five environments. We discuss the effect of each technique below.

**The effect of KL regularization** can be observed by comparing the results of KLENT and ENT Only. In ENT Only, KL regularization is removed so that the policy $\pi'$ is represented as $\pi'(a|s) = \frac{1}{Z(s)} \exp\left(Q_\theta(s,a)/\alpha\right)$. In other words, the output of the policy network is completely ignored, and actions are selected according to a softmax policy based solely on the action-value function. According to the results, ENT Only exhibits degraded performance compared to the original KLENT. Figure 8 shows the evolution of the average KL divergence $D_{\mathrm{KL}}(\pi'\|\pi)$ in 9x9 Go, where the performance gap between KLENT and ENT Only is large. While KLENT keeps the divergence relatively low via reverse-KL regularization, it is larger in ENT Only, suggesting more abrupt policy updates. These results suggest that it is important to gradually update the policy.

**The effect of entropy regularization** can be analyzed by comparing KLENT and KL Only. In KL Only, where the entropy regularization is removed, performance degrades significantly across all the five games. Specifically, in Animal Shogi, the win rate initially rises to 75% but subsequently declines, suggesting unstable learning. Figure 9 shows the evolution of the average entropy of the policy $\pi'$ in Animal Shogi. While KLENT maintains the entropy, it rapidly decreases and becomes nearly zero in KL Only, indicating that the policy becomes excessively deterministic. These results suggest that encouraging sufficient exploration is crucial for stable learning process.

**The effect of $\lambda$-returns** can be observed by comparing the results of KLENT, 1-Step KLENT, and Monte Carlo KLENT. Replacing $\lambda$-returns with 1-step returns or Monte Carlo returns results in a performance drop especially in 9x9 Go and Hex. As discussed in Section 4.2, the results suggest the importance of balancing bias-variance trade-off through the use of an intermediate $\lambda$.

## 6.3. Large-Scale Game

**Setup.** We further conducted experiments in 19x19 Go, comparing KLENT with AlphaZero. As Pgx (Koyamada et al., 2023) does not provide the pretrained checkpoint for 19x19 Go, we instead used the checkpoint released by ElfOpen Go (Tian et al., 2019) for the anchored opponent. For the network architecture, we used 20-block ResNet (He et al., 2016) instead of 6-block one to capture features in the larger board. KLENT used the same hyperparameters $(\alpha, \beta, \lambda) = (0.03, 0.1, e^{-1/8})$.

**Results.** We present the results in Figure 10. We can observe that even in 19x19 Go, KLENT achieves competitive learning compared to AlphaZero. Overall, our experimental results demonstrate that the KLENT achieves high learning efficiency in medium-scale environments, while also maintaining competitive learning in the large-scale environment.

## 7. Conclusions

In this study, we have revisited regularized policy optimization with reverse Kullback-Leibler regularization and entropy regularization and analyzed this combination on two-player zero-sum settings from theoretical and empirical perspectives. From theoretical perspective, we investigated the stability of the policy update rule on two theoretical settings: game-theoretic normal-form games and finite-length games. From empirical perspective, we have also investigated practical RL algorithm KLENT with this combination. Our experimental results have demonstrated learning efficiency of KLENT compared to existing methods. Through our ablation study, we also validated the importance of these three key techniques: KL regularization for gradual policy updates, entropy regularization for exploration, and $\lambda$-returns for efficient and stable value function learning.

A limitation of this study is that our empirical experiments are focused to improve the efficiency. While we have shown that KLENT can achieve efficient learning, this does not necessarily mean that it achieves superior asymptotic performance when unlimited computational resources are given. Although we consider our efficient approach to be valuable for most practitioners and researchers in the community, our results do not preclude the effectiveness of search-based approaches including AlphaZero, particularly when massive computational resources such as thousands of GPUs are available.

Our results have shown that even in board games, a domain long dominated by search-based methods, properly revisiting existing regularization techniques can lead to stable and efficient learning. We hope this encourages extending this approach to other fields where specialized methods prevail, further advancing reinforcement learning.

## Acknowledgements

We thank Yusuke Mukuta for fruitful discussions on the theoretical analysis. We thank Sotetsu Koyamada and Soichiro Nishimori for their kind guidance on using the Pgx library.

This work was partially supported by JST Moonshot R&D Grant Number JPMJPS2011, CREST Grant Number JPMJCR2015 and Basic Research Grant (Super AI) of Institute for AI and Beyond of the University of Tokyo. Kazuki Ota was supported by JST SPRING, Grant Number JPMJSP2108. Takayuki Osa was supported by JSPS KAKENHI Grant Number JP25K03176.

## Impact Statement

This paper presents work whose goal is to advance the field of Machine Learning. There are many potential societal consequences of our work, none of which we feel must be specifically highlighted here.

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

# Appendix

## Table of Contents

# Part I: Derivation of the Proposed Method

## A. Derivation of the Analytical Solution for the Regularized Policy Optimization Problem

Here, we provide the formal mathematical definitions of the terms in Definition A.1 and present the proof for the derivation of the optimal solution in Equation 3 in Theorem A.2. For simplicity, we do not explicitly write the considered state $s$ in the following equations.

**Definition A.1** (KL divergence and entropy). Let $\mathcal{A}$ be a finite set and $\Delta$ be the set of all probability mass functions over $\mathcal{A}$. The Kullback-Leibler (KL) divergence between two probability mass functions $\pi' \in \Delta$ and $\pi \in \Delta$ over a finite set $\mathcal{A}$ is defined as:

$$D_{\text{KL}}(\pi' \| \pi) = \sum_{a \in \mathcal{A}} \pi'(a) \log \frac{\pi'(a)}{\pi(a)}, \tag{7}$$

where it is assumed that $\pi'(a) = 0 \implies \pi'(a) \log \frac{\pi'(a)}{\pi(a)} = 0$ and $\pi(a) > 0$ for all $a \in \mathcal{A}$. The entropy of a probability mass function $\pi' \in \Delta$ over $\mathcal{A}$ is defined as:

$$H(\pi') = - \sum_{a \in \mathcal{A}} \pi'(a) \log \pi'(a), \tag{8}$$

where it is assumed that $\pi'(a) = 0 \implies \pi'(a) \log \pi'(a) = 0$.

**Theorem A.2** (Formal Derivation of the Analytical Solution $\pi'$). *Let $\mathcal{A}$ be a finite set, $\pi(a)$ a probability mass function over $\mathcal{A}$, $Q(a) : \mathcal{A} \to \mathbb{R}$ a function, and $\Delta$ the set of all probability mass functions over $\mathcal{A}$. Consider the following optimization problem:*

$$\underset{\pi' \in \Delta}{\text{maximize}} \ \mathbb{E}_{A \sim \pi'}[Q(A)] - \beta D_{KL}(\pi' \| \pi) + \alpha H(\pi'), \tag{9}$$

*where $\beta > 0$ and $\alpha > 0$. Then, the optimal solution is given by:*

$$\pi'(a) = \frac{1}{Z} \exp \left( \frac{Q(a) + \beta \log \pi(a)}{\alpha + \beta} \right), \tag{10}$$

*where*

$$Z = \sum_{a \in \mathcal{A}} \exp \left( \frac{Q(a) + \beta \log \pi(a)}{\alpha + \beta} \right) \tag{11}$$

*is the normalization constant.*

*Proof.* Define the Lagrangian as follows:

$$\mathcal{L}(\pi', \lambda) = \sum_{a \in \mathcal{A}} \pi'(a) Q(a) - \beta D_{\text{KL}}(\pi' \| \pi) + \alpha H(\pi') - \lambda \left( \sum_{a \in \mathcal{A}} \pi'(a) - 1 \right), \tag{12}$$

where $\lambda$ is the Lagrange multiplier enforcing the constraint that $\pi'(a)$ is a probability mass function.

Using the method of Lagrange multipliers, we find $\pi'$ that satisfies

$$\nabla_{\pi'} \mathcal{L}(\pi', \lambda) = 0. \tag{13}$$

Expanding this condition yields:

$$\nabla_{\pi'} \mathcal{L}(\pi', \lambda) = 0 \tag{14}$$

$$\iff \nabla_{\pi'} \left( \sum_{a \in \mathcal{A}} \pi'(a) Q(a) - \beta D_{\text{KL}}(\pi' \| \pi) + \alpha H(\pi') - \lambda \left( \sum_{a \in \mathcal{A}} \pi'(a) - 1 \right) \right) = 0 \tag{15}$$

$$\iff Q(a) + \beta \log \pi(a) - (\beta + \alpha)(\log \pi'(a) + 1) - \lambda = 0, \quad \forall a \in \mathcal{A} \tag{16}$$

$$\iff \log \pi'(a) = \frac{Q(a) + \beta \log \pi(a)}{\beta + \alpha} + (\text{const.}), \quad \forall a \in \mathcal{A}. \tag{17}$$

Since $\pi'(a)$ must be a probability mass function, the solution is given by Equation 10. $\square$

*Remark* A.3. Here, the Lagrange multiplier $\lambda$ is unrelated to the $\lambda$ in $\lambda$-returns.

*Remark* A.4. The advantage of combining reverse KL and entropy regularization is that it yields an explicit, closed-form analytical update. For example, forward KL regularization alone admits an analytical solution (Grill et al., 2020), but it lacks explicit exploration. Conversely, combining forward KL and entropy leads to an optimality equation involving the Lambert W function, and does not yield an elementary closed-form update without special functions (Chow, 1999). In contrast, combining reverse KL and entropy provides a directly computable analytical solution, as shown in Theorem A.2.

## B. Details of Preliminary Experiments on Bias-Variance Trade-Off

In Figure 2, we have demonstrated the bias-variance trade-off of $\lambda$-returns in 9x9 Go environment. For the detailed experimental setup, we fixed both the policy and the value function using a pretrained baseline model from Pgx library (Koyamada et al., 2023) in order to isolate the effect of varying $\lambda$. Since estimating the bias requires access to the true action value, which is not directly observable, we approximated the ground-truth value by computing the Monte Carlo return 1,000 times from the same state action pair and taking the average as a surrogate for the true value.

# Part II: Details of Theoretical Analysis

## C. Details of Theoretical Analysis

### C.1. Normal-Form Games

In this subsection, we analyze the local linear convergence of the regularized policy update map that arises in two-player zero-sum normal-form games.

**Definition C.1** (Simplex and its interior). For a positive integer $m$, define

$$\Delta := \{p \in \mathbb{R}^m \mid \forall i,\ p_i \geq 0,\ \langle p, \mathbf{1} \rangle = 1\}, \qquad \mathrm{int}\Delta := \{p \in \Delta \mid \forall i,\ p_i > 0\},$$

where $\mathbf{1} = (1, \ldots, 1)^\top \in \mathbb{R}^m$.

**Definition C.2** (Component-wise logarithm). For $p \in \mathrm{int}\Delta$, define

$$\log p := (\log p_1, \ldots, \log p_m)^\top \in \mathbb{R}^m.$$

**Definition C.3** (Entropy). Define $H : \mathrm{int}\Delta \to \mathbb{R}$ by

$$H(p) := -p^\top \log p = -\sum_{i=1}^m p_i \log p_i.$$

When needed, we also use its continuous extension to $\Delta$:

$$H(p) := -\sum_{i=1}^m p_i \log p_i, \qquad 0 \log 0 := 0.$$

**Definition C.4** (Softmax). Define $\sigma : \mathbb{R}^m \to \mathrm{int}\Delta$ by

$$\forall z = (z_1, \ldots, z_m)^\top \in \mathbb{R}^m, \quad \sigma(z) := \left( \frac{e^{z_1}}{\sum_{j=1}^m e^{z_j}}, \ldots, \frac{e^{z_m}}{\sum_{j=1}^m e^{z_j}} \right)^\top.$$

Softmax is invariant to constant shifts: for any $\kappa \in \mathbb{R}$,

$$\sigma(z + \kappa \mathbf{1}) = \sigma(z).$$

**Lemma C.5** (Jacobian of softmax). *For any $x \in \mathbb{R}^m$,*

$$\nabla \sigma(x) = \mathrm{diag}(\sigma(x)) - \sigma(x)\sigma(x)^\top.$$

*Moreover, if $p = \sigma(x) \in \mathrm{int}\Delta$, then $\mathrm{diag}(p) - pp^\top$ is positive semidefinite with $\mathrm{Ker}(\mathrm{diag}(p) - pp^\top) = \mathrm{span}\{\mathbf{1}\}$, and hence it is positive definite on $\mathbf{1}^\perp := \{v \in \mathbb{R}^m \mid \langle v, \mathbf{1} \rangle = 0\}$.*

**Lemma C.6** (Operator-norm bound of the softmax Jacobian). *For any $p \in \Delta$,*

$$\left\| \mathrm{diag}(p) - pp^\top \right\|_2 \leq \frac{1}{2}.$$

*Consequently, for any $z \in \mathbb{R}^m$,*

$$\|\nabla \sigma(z)\|_2 \leq \frac{1}{2}.$$

*Proof.* Let $v \in \mathbb{R}^m$ be any unit vector. Then

$$v^\top (\mathrm{diag}(p) - pp^\top)v = \sum_{i=1}^m p_i v_i^2 - \left(\sum_{i=1}^m p_i v_i\right)^2$$

equals the variance $\mathrm{Var}(X)$ of a random variable $X$ with $\mathbb{P}(X = v_i) = p_i$. By Popoviciu's inequality,

$$\mathrm{Var}(X) \leq \frac{(\max X - \min X)^2}{4}.$$

Since $\max_i v_i - \min_j v_j \leq \max_{i,j} |v_i - v_j| \leq \max_{i,j} \|e_i - e_j\|_2 \|v\|_2 = \sqrt{2}$, we obtain $\mathrm{Var}(X) \leq (\sqrt{2})^2/4 = 1/2$. Taking the supremum over unit vectors $v$ yields $\|\mathrm{diag}(p) - pp^\top\|_2 \leq 1/2$. The second claim follows from Lemma C.5 with $p = \sigma(z)$. $\qquad\square$

**Lemma C.7** (Local linear convergence from Jacobian contraction). *Let $(\mathbb{R}^n, \|\cdot\|)$ be a normed space and let $F : \mathbb{R}^n \to \mathbb{R}^n$ be $C^1$. Assume $x^* \in \mathbb{R}^n$ satisfies $F(x^*) = x^*$ and $\|\nabla F(x^*)\|_{\mathrm{op}} < 1$. Then there exist $\varepsilon > 0$ and $q \in (0,1)$ such that for all $x$ with $\|x - x^*\| < \varepsilon$ and all $k \geq 0$,*

$$\|F^k(x) - x^*\| \leq q^k \|x - x^*\|.$$

*In other words, $F^k(x)$ converges to $x^*$ locally linearly.*

*Proof.* By continuity of $\nabla F$, there exist $\varepsilon > 0$ and $q \in (0,1)$ such that $\|\nabla F(x)\|_{\mathrm{op}} \leq q$ for all $x$ in $B := \{x \mid \|x - x^*\| < \varepsilon\}$. For any $x, y \in B$, define $\gamma(t) = tx + (1 - t)y$. By the fundamental theorem of calculus,

$$F(x) - F(y) = \int_0^1 \nabla F(\gamma(t))(x - y)\, dt,$$

hence $\|F(x) - F(y)\| \leq \int_0^1 \|\nabla F(\gamma(t))\|_{\mathrm{op}}\, dt \cdot \|x - y\| \leq q\|x - y\|$. Thus $F$ is a contraction on $B$. Taking $y = x^*$ and using $F(x^*) = x^*$, we obtain

$$\|F(x) - x^*\| \leq q\|x - x^*\| \qquad (x \in B).$$

Hence $F(B) \subset B$. By induction, for all $k \geq 0$ and $x \in B$,

$$\|F^k(x) - x^*\| \leq q^k \|x - x^*\|.$$

This inequality gives local linear convergence. In particular, since $q \in (0,1)$, we have $q^k \to 0$, and therefore $\lim_{k \to \infty} F^k(x) = x^*$. $\qquad\square$

**Theorem C.8** (Local linear convergence of the regularized dynamics). *Let $\alpha, \beta \in \mathbb{R}_{>0}$ and let $A \in \mathbb{R}^{m \times m}$. Define $f : (\mathrm{int}\Delta)^2 \to (\mathrm{int}\Delta)^2$ by*

$$f\left(\begin{pmatrix} p \\ q \end{pmatrix}\right) = \begin{pmatrix} \sigma\left(\frac{Aq + \beta \log p}{\alpha + \beta}\right) \\ \sigma\left(\frac{-A^\top p + \beta \log q}{\alpha + \beta}\right) \end{pmatrix}.$$

*Let $(p^*, q^*) \in (\mathrm{int}\Delta)^2$ be a fixed point of $f$. Define*

$$P^* := \mathrm{diag}(p^*) - p^* p^{*\top}, \qquad Q^* := \mathrm{diag}(q^*) - q^* q^{*\top}.$$

*If*

$$\alpha(\alpha + 2\beta) > \|P^{*1/2} A Q^{*1/2}\|_2^2, \tag{18}$$

*then $f$ is locally linearly convergent around $(p^*, q^*)$, which implies that there exists a neighborhood $U$ of $(p^*, q^*)$ such that for all $(p_0, q_0) \in U$, the iterates $(p_{k+1}, q_{k+1}) = f(p_k, q_k)$ satisfy*

$$\lim_{k \to \infty} (p_k, q_k) = (p^*, q^*).$$

*Proof.* Step 1: Coordinate transform. Let

$$\Pi := I - \frac{1}{m}\mathbf{1}\mathbf{1}^\top$$

be the orthogonal projection onto $\mathbf{1}^\perp$. By Definition C.4, softmax is shift-invariant, hence for any $z \in \mathbb{R}^m$,

$$\sigma(z) = \sigma(\Pi z).$$

Define $F : \mathbf{1}^\perp \times \mathbf{1}^\perp \to \mathbf{1}^\perp \times \mathbf{1}^\perp$ by

$$F\left(\begin{pmatrix} x \\ y \end{pmatrix}\right) := \begin{pmatrix} \Pi\frac{A\sigma(y)+\beta x}{\alpha+\beta} \\ \Pi\frac{-A^\top\sigma(x)+\beta y}{\alpha+\beta} \end{pmatrix} = \frac{1}{\alpha+\beta}\begin{pmatrix} \Pi & 0 \\ 0 & \Pi \end{pmatrix}\begin{pmatrix} A\sigma(y) + \beta x \\ -A^\top\sigma(x) + \beta y \end{pmatrix}.$$

Using $\sigma(\cdot) = \sigma(\Pi\cdot)$, we can write

$$f = \begin{pmatrix} \sigma \\ \sigma \end{pmatrix} \circ F \circ \begin{pmatrix} \log \\ \log \end{pmatrix},$$

where $\begin{pmatrix} \log \\ \log \end{pmatrix}(p,q) = (\log p, \log q)$. Therefore, local convergence of $F$ around the fixed point

$$(x^*, y^*) := (\Pi \log p^*,\ \Pi \log q^*)$$

implies local convergence of $f$ around $(p^*, q^*)$ by continuity of $\log$ and $\sigma$.

Step 2: Jacobian of $F$ at the fixed point. Let $P^* = \nabla\sigma(x^*)$ and $Q^* = \nabla\sigma(y^*)$. By Lemma C.5, these coincide with $P^* = \mathrm{diag}(p^*) - p^*p^{*\top}$ and $Q^* = \mathrm{diag}(q^*) - q^*q^{*\top}$. Differentiating $F$ yields

$$\nabla F(x^*, y^*) = \frac{1}{\alpha+\beta}\begin{pmatrix} \Pi & 0 \\ 0 & \Pi \end{pmatrix}\begin{pmatrix} \beta I & AQ^* \\ -A^\top P^* & \beta I \end{pmatrix}.$$

Since the domain is $\mathbf{1}^\perp \times \mathbf{1}^\perp$, $\Pi$ acts as the identity and can be omitted when evaluating operator norms restricted to this subspace.

Step 3: A weighted norm and the operator-norm bound. Define a norm $\|\cdot\|_* : \mathbf{1}^\perp \times \mathbf{1}^\perp \to \mathbb{R}_{\geq 0}$ by

$$\left\|\begin{pmatrix} x \\ y \end{pmatrix}\right\|_* := \sqrt{x^\top P^* x + y^\top Q^* y}.$$

By Lemma C.5, $P^*$ and $Q^*$ are symmetric positive definite on $\mathbf{1}^\perp$, so $\|\cdot\|_*$ is a valid norm on $\mathbf{1}^\perp \times \mathbf{1}^\perp$. Let $S := \mathrm{diag}(P^{*1/2}, Q^{*1/2})$. Then for $z = \begin{pmatrix} x \\ y \end{pmatrix}$,

$$\|z\|_* = \|Sz\|_2.$$

Hence for any linear map $L$ on $\mathbf{1}^\perp \times \mathbf{1}^\perp$,

$$\|L\|_{*,\mathrm{op}} = \|SLS^{-1}\|_2.$$

Applying this to $L = \nabla F(x^*, y^*)$ gives

$$\|\nabla F(x^*, y^*)\|_{*,\mathrm{op}} = \frac{1}{\alpha+\beta}\left\|\begin{pmatrix} \beta I & B \\ -B^\top & \beta I \end{pmatrix}\right\|_2, \qquad B := P^{*1/2}AQ^{*1/2}.$$

Let $M := \begin{pmatrix} \beta I & B \\ -B^\top & \beta I \end{pmatrix}$. Then

$$M^\top M = \begin{pmatrix} \beta^2 I + BB^\top & 0 \\ 0 & \beta^2 I + B^\top B \end{pmatrix},$$

and therefore

$$\|M\|_2^2 = \|M^\top M\|_2 = \beta^2 + \|B\|_2^2 \quad \Longrightarrow \quad \|M\|_2 = \sqrt{\beta^2 + \|B\|_2^2}.$$

Consequently,
$$\|\nabla F(x^*, y^*)\|_{*,\mathrm{op}} = \frac{1}{\alpha + \beta}\sqrt{\beta^2 + \|B\|_2^2}.$$

Thus $\|\nabla F(x^*, y^*)\|_{*,\mathrm{op}} < 1$ is equivalent to
$$\beta^2 + \|B\|_2^2 < (\alpha + \beta)^2 \quad \Longleftrightarrow \quad \|B\|_2^2 < \alpha(\alpha + 2\beta),$$

which is exactly (18). By Lemma C.7, $F$ is locally linearly convergent around $(x^*, y^*)$ under $\|\cdot\|_*$. Moreover, $\log$ and $\sigma$ are locally Lipschitz around $(p^*, q^*)$ and $(x^*, y^*)$, respectively. Hence, by the composition identity $f = \begin{pmatrix} \sigma \\ \sigma \end{pmatrix} \circ F \circ \begin{pmatrix} \log \\ \log \end{pmatrix}$, $f$ is locally linearly convergent around $(p^*, q^*)$. $\qquad\square$

**Corollary C.9.** *Under the same definitions as in Theorem C.8, the regularized policy update rule is locally linearly convergent to the unique fixed point if the following condition is satisfied:*

$$\alpha(\alpha + 2\beta) > \frac{1}{4}\|A\|_2^2. \tag{19}$$

*Proof.* Recall the definition $B = P^{*1/2}AQ^{*1/2}$ from the proof of Theorem C.8. Using the submultiplicativity of the spectral norm, we have
$$\|B\|_2 \le \|P^{*1/2}\|_2\|A\|_2\|Q^{*1/2}\|_2.$$
Note that for any positive semidefinite matrix $M$, $\|M^{1/2}\|_2 = \sqrt{\|M\|_2}$. From Lemma C.6, we have the bounds $\|P^*\|_2 \le 1/2$ and $\|Q^*\|_2 \le 1/2$. Substituting these into the inequality yields
$$\|B\|_2 \le \sqrt{\frac{1}{2}}\|A\|_2\sqrt{\frac{1}{2}} = \frac{1}{2}\|A\|_2.$$

Squaring both sides, we obtain
$$\|B\|_2^2 \le \frac{1}{4}\|A\|_2^2.$$

Therefore, the condition provided in the corollary,
$$\frac{1}{4}\|A\|_2^2 < \alpha(\alpha + 2\beta),$$

is sufficient to guarantee
$$\|B\|_2^2 < \alpha(\alpha + 2\beta),$$

which satisfies the sufficient condition for local linear convergence (18) in Theorem C.8. $\qquad\square$

Theorem C.8 extends to general-sum settings as follows.

**Theorem C.10** (General-sum extension of local linear convergence). *Let $\alpha, \beta \in \mathbb{R}_{>0}$, $A_1, A_2 \in \mathbb{R}^{m\times n}$, and define*

$$f : \big(\mathrm{int}\Delta_m\big) \times \big(\mathrm{int}\Delta_n\big) \to \big(\mathrm{int}\Delta_m\big) \times \big(\mathrm{int}\Delta_n\big)$$

*by*

$$f\left(\begin{pmatrix} p \\ q \end{pmatrix}\right) = \begin{pmatrix} \sigma\left(\dfrac{A_1 q + \beta \log p}{\alpha + \beta}\right) \\ \sigma\left(\dfrac{A_2^\top p + \beta \log q}{\alpha + \beta}\right) \end{pmatrix}.$$

*Let $(p^*, q^*) \in (\mathrm{int}\Delta_m) \times (\mathrm{int}\Delta_n)$ be a fixed point of $f$, and define*

$$P^* := \mathrm{diag}(p^*) - p^*p^{*\top}, \qquad Q^* := \mathrm{diag}(q^*) - q^*q^{*\top},$$
$$B_1 := P^{*1/2}A_1Q^{*1/2}, \qquad B_2 := Q^{*1/2}A_2^\top P^{*1/2}.$$

*If*

$$\left\|\begin{pmatrix} \beta I_m & B_1 \\ B_2 & \beta I_n \end{pmatrix}\right\|_2 < \alpha + \beta,$$

*then $f$ is locally linearly convergent around $(p^*, q^*)$.*

*Proof.* Let

$$\Pi_m := I_m - \frac{1}{m}\mathbf{1}_m\mathbf{1}_m^\top, \qquad \Pi_n := I_n - \frac{1}{n}\mathbf{1}_n\mathbf{1}_n^\top,$$

and define $F : \mathbf{1}_m^\perp \times \mathbf{1}_n^\perp \to \mathbf{1}_m^\perp \times \mathbf{1}_n^\perp$ by

$$F\left(\begin{pmatrix} x \\ y \end{pmatrix}\right) := \begin{pmatrix} \Pi_m \dfrac{A_1\sigma(y) + \beta x}{\alpha + \beta} \\ \Pi_n \dfrac{A_2^\top \sigma(x) + \beta y}{\alpha + \beta} \end{pmatrix}.$$

With $(x^*, y^*) := (\Pi_m \log p^*, \Pi_n \log q^*)$, we have $\sigma(x^*) = p^*$, $\sigma(y^*) = q^*$, and $(x^*, y^*)$ is a fixed point of $F$. Moreover, by shift invariance of softmax,

$$f = \begin{pmatrix} \sigma \\ \sigma \end{pmatrix} \circ F \circ \begin{pmatrix} \Pi_m \log \\ \Pi_n \log \end{pmatrix}.$$

At $(x^*, y^*)$, using $\nabla\sigma(x^*) = P^*$ and $\nabla\sigma(y^*) = Q^*$,

$$\nabla F(x^*, y^*) = \frac{1}{\alpha + \beta} \begin{pmatrix} \beta I_m & \Pi_m A_1 Q^* \\ \Pi_n A_2^\top P^* & \beta I_n \end{pmatrix}.$$

On $\mathbf{1}_m^\perp \times \mathbf{1}_n^\perp$, $\Pi_m, \Pi_n$ act as identities, so for $z = (x, y)$ in this subspace,

$$\nabla F(x^*, y^*)z = \frac{1}{\alpha + \beta} \begin{pmatrix} \beta x + A_1 Q^* y \\ A_2^\top P^* x + \beta y \end{pmatrix}.$$

Define the weighted norm

$$\left\| \begin{pmatrix} x \\ y \end{pmatrix} \right\|_* := \sqrt{x^\top P^* x + y^\top Q^* y}.$$

Since $P^*, Q^*$ are positive definite on $\mathbf{1}_m^\perp, \mathbf{1}_n^\perp$, this is a norm. Let $u := P^{*1/2}x$, $v := Q^{*1/2}y$. Then

$$\|z\|_* = \left\| \begin{pmatrix} u \\ v \end{pmatrix} \right\|_2$$

and

$$\|\nabla F(x^*, y^*)z\|_* = \frac{1}{\alpha + \beta} \left\| \begin{pmatrix} \beta I_m & B_1 \\ B_2 & \beta I_n \end{pmatrix} \begin{pmatrix} u \\ v \end{pmatrix} \right\|_2.$$

Hence

$$\|\nabla F(x^*, y^*)\|_{*,\mathrm{op}} = \frac{1}{\alpha + \beta} \left\| \begin{pmatrix} \beta I_m & B_1 \\ B_2 & \beta I_n \end{pmatrix} \right\|_2 < 1.$$

By Lemma C.7, $F$ is locally linearly convergent around $(x^*, y^*)$: there exist $\varepsilon > 0$, $r \in (0, 1)$ such that

$$\|F^k(z) - z^*\|_* \le r^k \|z - z^*\|_*$$

for all $z$ with $\|z - z^*\|_* < \varepsilon$, where $z^* := (x^*, y^*)$.

Finally, $(\Pi_m \log, \Pi_n \log)$ and $(\sigma, \sigma)$ are locally Lipschitz around $(p^*, q^*)$ and $(x^*, y^*)$, respectively. Therefore, there exist constants $C_1, C_2 > 0$ and a neighborhood $U$ of $(p^*, q^*)$ such that for all $(p_0, q_0) \in U$, Since $(\Pi_m \log, \Pi_n \log) \circ (\sigma, \sigma) = \mathrm{id}$ on $\mathbf{1}_m^\perp \times \mathbf{1}_n^\perp$, we have $f^k = (\sigma, \sigma) \circ F^k \circ (\Pi_m \log, \Pi_n \log)$ for all $k \ge 0$.

$$\left\| f^k(p_0, q_0) - (p^*, q^*) \right\|_2 \le C_2 \, r^k \, C_1 \, \|(p_0, q_0) - (p^*, q^*)\|_2.$$

Thus $f$ is locally linearly convergent around $(p^*, q^*)$. $\qquad\square$

*Remark* C.11. Extending Theorem C.8, Theorem C.10, and Corollary C.9 to multi-player games is challenging. Two-player games use a two-dimensional payoff matrix, whereas three or more players require a higher-order payoff tensor. The contraction-mapping approach may remain effective, but analyzing the higher-order mapping requires extending the current proof framework beyond the Jacobian-based block-matrix analysis used here.

## C.2. Finite-Length Games

**Theorem C.12** (Convergence of KLENT under bounded episode length). *Assume that there exists $T_{\max}$ such that the terminal step $T$ always satisfies $T \leq T_{\max}$. Then the KLENT policy converges to an entropy-regularized optimal policy that satisfies*

$$\pi(a|s) \propto \exp\left(\frac{Q^\pi(s,a)}{\alpha}\right), \qquad \alpha > 0.$$

*Proof.* We use mathematical induction together with a multi-stage convergence argument. Assign to each state $s$ an index defined as "the maximum number of steps required to reach a terminal state from $s$." Because we assume the existence of $T_{\max}$, this index can be assigned to every state. For each state $s$, let $\pi_n(\cdot|s)$ denote the policy after the $n$-th update.

Step 1. Base case: index $= 1$: If the index of $s$ is 1, then from $s$ the next state must be terminal. Hence the value of the next state is fully determined by the environment reward, and, after sufficiently many updates, the action-value estimate at $s$ converges to that value. In particular, after sufficiently many updates, $Q_\theta(s,a)$ has converged to $Q^\pi(s,a)$ for each action $a$. Fix a reference action $a_{\mathrm{ref}}$ with $\pi_n(a_{\mathrm{ref}}|s) > 0$ and define

$$z_n(a) := \log \frac{\pi_n(a|s)}{\pi_n(a_{\mathrm{ref}}|s)}.$$

Taking the log-ratio of Equation 3 for $a$ and $a_{\mathrm{ref}}$ gives

$$\log \frac{\pi_{n+1}(a|s)}{\pi_{n+1}(a_{\mathrm{ref}}|s)} = \log \frac{\frac{1}{Z_n(s)} \exp\left(\frac{Q^\pi(s,a) + \beta \log \pi_n(a|s)}{\alpha + \beta}\right)}{\frac{1}{Z_n(s)} \exp\left(\frac{Q^\pi(s,a_{\mathrm{ref}}) + \beta \log \pi_n(a_{\mathrm{ref}}|s)}{\alpha + \beta}\right)}$$

$$= \frac{Q^\pi(s,a) - Q^\pi(s,a_{\mathrm{ref}}) + \beta(\log \pi_n(a|s) - \log \pi_n(a_{\mathrm{ref}}|s))}{\alpha + \beta}.$$

By the definition of $z_n(a)$, this is equivalent to

$$z_{n+1}(a) = \frac{\beta}{\alpha + \beta} z_n(a) + \frac{Q^\pi(s,a) - Q^\pi(s,a_{\mathrm{ref}})}{\alpha + \beta},$$

whose explicit solution is

$$z_n(a) = z_*(a) + \left(\frac{\beta}{\alpha + \beta}\right)^n (z_0(a) - z_*(a)), \qquad z_*(a) = \frac{Q^\pi(s,a) - Q^\pi(s,a_{\mathrm{ref}})}{\alpha}.$$

Therefore $z_n(a) \to z_*(a)$, and after normalization $\pi_n(\cdot|s)$ converges to the desired Boltzmann policy.

Step 2. Induction step: Assume that, for all states whose indices are at most $k$, both the policy and the action-value estimates have converged. Consider a state whose index is $k + 1$. By the same argument as in the base case, after sufficiently many updates, the action-value estimate at that state converges to $Q^\pi$, and the same logit-recurrence argument implies that the policy at that state converges to the desired Boltzmann form.

By induction, the claim holds for all states. $\qquad\square$

*Remark* C.13. The proof of Theorem C.12 relies primarily on a bounded horizon and backward induction, and depends less on the two-player zero-sum structure. This suggests that the argument may naturally extend to multi-player general-sum settings.

*Remark* C.14. Our current proof for finite-horizon games is based on backward induction and relies on the existence of $T_{\max}$, so it does not directly extend to infinite-horizon games. Therefore, a theoretical extension to infinite-horizon games would likely require a different analytical framework, for example one incorporating discounting or a stochastic horizon.

# D. Details on Count Up Game

In this section, we report the details of experiments on count up game. In two-player games, quantal response equilibrium (McKelvey & Palfrey, 1995) is defined as a policy which satisfies the following equation:

$$\pi(a|s) = \frac{1}{Z(s)} \exp(Q^\pi(s,a)/\alpha).$$

Sokota et al. (2022) have provided a theoretical analysis on the combination of KL and entropy regularization, and it suggests that the convergence limit of this combination is the quantal response equilibrium. To verify this expectation, we have conducted experiments on a simple small-scale game, namely, the count-up game. The rule is defined as follows.

**Formal Rule**   Let us consider a two-player sequential zero-sum game. Let $N$ and $k$ be positive integers. The state space is non-negative integers $\mathcal{S} = \{0, 1, 2, \dots\}$ and the action space is positive integers up to $k$: $\mathcal{A} = \{1, \dots, k\}$. The initial state $S_0$ is always 0, and the state transition, termination, and rewards are defined as follows.

- Transition: The next state is defined as $S_{t+1} = S_t + A_t$.

- Termination: The game terminates when $S_t + A_t \geq N$.

- Rewards: The reward is defined as follows:

$$r(S_t, A_t) = \begin{cases} 1 & \text{if } S_t + A_t \geq N \\ 0 & \text{otherwise} \end{cases}$$

**Interpretation of the Rule**   This rule can be interpreted as follows. There are two players and they declare the number of $S_{t+1}$ alternately. Let $N = 7$ and $k = 2$, for example. Then, the first player can declare 1 or 2, and the next player can declare a number that is larger by 1 or 2 than the previously declared number. If a player declares a number which is equal to or larger than 7, the player wins the game.

In this simple and small-scale game, we analyze the convergence limit of KLENT. Below, we assume that $N = 7$ and $k = 2$ unless otherwise described.

**Optimal Strategy**   The optimal strategy of this game can be calculated in a backward manner as Table 3.

*Table 3.* Optimal strategy in the count up game.

| State $S_t$ | Optimal Strategy |
|:---:|:---:|
| 6 | Win with $A_t = +1$ or $+2$. |
| 5 | Win with $A_t = +2$. |
| 4 | Lose anyway. |
| 3 | Win with $A_t = +1$. |
| 2 | Win with $A_t = +2$. |
| 1 | Lose anyway. |
| 0 | Win with $A_t = +1$. |

**Quantal Response Equilibrium**   In this game, quantal response equilibrium can also be calculated in a backward manner. We have calculated them for $\alpha = 0.03$ and $\alpha = 1.0$. The results are shown in Figure 11. It can be observed that the equilibrium of $\alpha = 0.03$ is close to the optimal strategy in Table 3, and that of $\alpha = 1.0$ is a softer policy.

**Convergence Limit of KLENT**   We have run the KLENT algorithm on this game, especially with $\alpha = 1.0$. The evolution of learned policy and action values is shown in Figure 12. The results confirm the expectation that KLENT converges to the quantal response equilibrium.

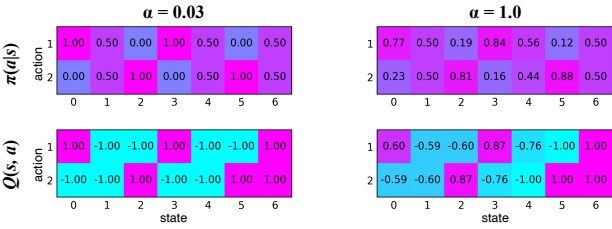

*Figure 11.* Quantal response equilibrium in the count-up game.

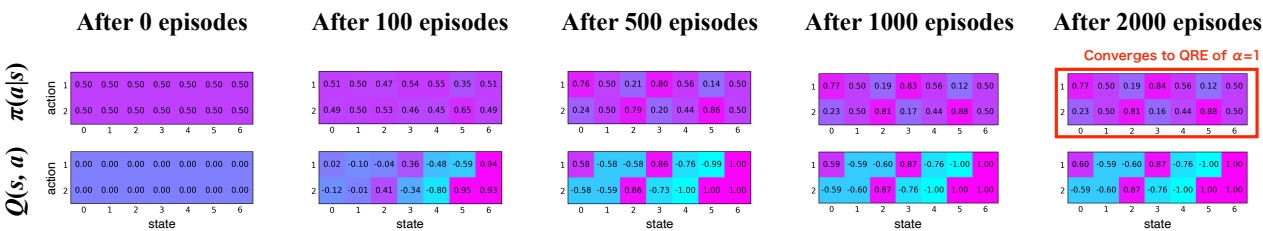

*Figure 12.* The evolution of learned policy and action values of KLENT in the count-up game. The results show that KLENT converges to the quantal response equilibrium.

# Part III: Details of Experiments

## E. Experimental Setup Details

We explain the detailed experimental setup in this section. For performance evaluation, we used the baseline opponent provided by Pgx as an anchored opponent. This anchored opponent selects actions stochastically based on its policy. The evaluated methods used deterministic policies by setting the temperature parameter to zero for softmax policies and $\epsilon$ to zero for $\epsilon$-greedy policies. In particular, the KLENT uses the greedy policy corresponding to the output $\pi$ of the policy network. In the evaluation, all agents select actions without search unifying their test-time computational resources[2]. The evaluation was conducted by playing 1024 matches against the anchored opponent, and the win rate was plotted on the vertical axis of the graph. Draws were treated as half-wins. The horizontal axis represents the total number of simulator evaluations during training, which includes all interactions with the environment simulator such as rollouts in tree search. This choice is consistent with prior literature that measures computational cost in terms of simulator evaluations, as seen in studies such as KataGo (Wu et al., 2020). Methods closer to the upper-left in the graph are interpreted as more efficient, achieving higher performance with fewer simulator accesses. For each method, experiments were conducted using three random seeds, and the mean and standard deviation of the obtained metrics were displayed on the graph.

## F. Implementation Details

This section describes the implementation details used in the experiments. For model-based methods, AlphaZero, TRPO AlphaZero, and Gumbel AlphaZero, we used open-source implementations provided by Mctx (Danihelka et al., 2022) and Pgx (Koyamada et al., 2023). Each iteration performed self-play in parallel across 1024 threads, with each thread executing up to 256 state transitions. If a game ended before 256 steps, a new game state was immediately initialized to continue the threads. Monte Carlo tree search was conducted for decision-making with a simulation budget of 32 for each action selection.

For model-free methods, including PPO, DQN, and our agent KLENT, self-play was similarly conducted in parallel across 1024 threads, but with each thread executing up to 2048 state transitions without search. The process for initializing new

---

[2]For search-based evaluations, please refer to Appendix M and N.

games upon completion was the same as for model-based methods. The hyperparameters of our agent KLENT were set as $(\alpha, \beta, \lambda) = (0.03, 0.1, e^{-1/8})$, as specified in Appendix E. The hyperparameters for PPO and DQN were determined referring to the implementation in Stable-Baselines3 (Raffin et al., 2021). For PPO, the regularization applied a clipping method to impose proximity, with the clipping ratio set to 0.2. The Generalized Advantage Estimator (GAE) in PPO used the same $\lambda = e^{-1/8}$ as KLENT. In the case of DQN, the $\epsilon$-greedy policy started with an $\epsilon$ value of 1.0, which was linearly reduced to 0.05 over the first $10^8$ simulator evaluations, and fixed at 0.05 thereafter.

The network architecture was consistent across all methods and based on ResNetV2 (He et al., 2016). The number of hidden layer channels was set to 128, for 6 residual blocks. Policy, state-value, and action-value heads were added as required by each method. Table 4 summarizes the inclusion of these heads for each method. The network takes a state observation as an input, with the policy head and action-value head outputting $|\mathcal{A}|$-dimensional vectors, and the state-value head outputting a scalar value. Due to variations in input and output shapes depending on the games and methods, the number of parameters varied slightly but remained within the range of 1.7 to 2.1 million across all experimental settings. Training of the networks was conducted with a batch size of 4096, a learning rate of 0.001, and the Adam optimizer (Kingma & Ba, 2015).

*Table 4.* Summary of the network heads included for each method.

|  | Policy Head | State-Value Head | Action-Value Head |
|---|---|---|---|
| KLENT | Yes | No | Yes |
| AlphaZero | Yes | Yes | No |
| TRPO AlphaZero | Yes | Yes | No |
| Gumbel AlphaZero | Yes | Yes | No |
| DQN | No | No | Yes |
| PPO | Yes | Yes | No |

## G. Sensitivity Analysis of Hyperparameters in KLENT

This section examines the performance variation of KLENT with respect to changes in the hyperparameters $\alpha, \beta, \lambda$. Specifically, for 9x9 Go, we conducted experiments on 27 combinations of hyperparameter values as follows: $(\alpha, \beta, \lambda) \in \{0.01, 0.03, 0.1\} \times \{0.03, 0.1, 0.3\} \times \{e^{-1/4}, e^{-1/8}, e^{-1/16}\}$. For each combination, we used three random seeds and calculated the average win rate against the anchored opponent during the training steps between 600 and 800 million simulator evaluations. The results are shown in Figure 13. Other experimental settings follow those described in Section 6.

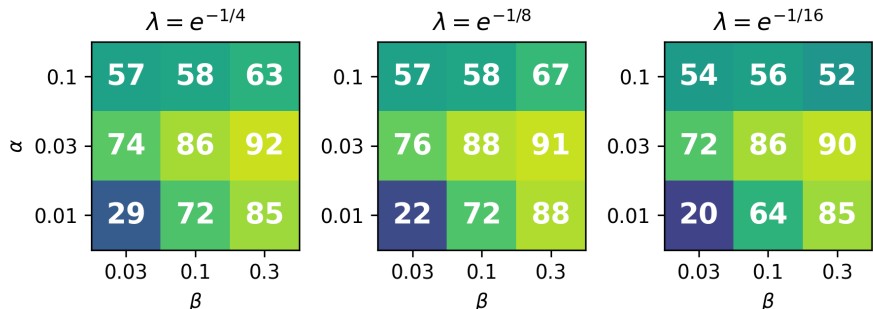

*Figure 13.* The results of sensitivity analysis in 9x9 Go.

When the coefficients of KL regularization and entropy regularization were both set to small values, specifically $(\alpha, \beta) = (0.01, 0.03)$, a notable decline in performance was observed. This is likely due to the improved policy, defined by Equation 3, becoming overly sharp. These results suggest that the regularization coefficients need to be set to sufficiently large and appropriate values. On the other hand, within the range of experiments conducted, the performance appears to be robust to variations in the time constant of $\lambda$-returns.

## H. Experimental Comparison of Network Architectures

In the main paper, we used the AlphaZero-style shared-backbone separate-heads architecture throughout in order to compare RL algorithms under a common setup. However, we agree that the reviewer's question is important, and we therefore conducted additional experiments on 9x9 Go with the following two architectures.

- (i) shared-backbone single-head: a single-network variant where policy and action-value fully share one output representation, with both training losses applied to that shared head.

- (ii) separate-backbones separate-heads (2x Computation Cost): a fully separated two-network variant with independent backbones and heads for policy and action-value, so that no internal representation is shared.

All other configurations were kept identical to those of the original KLENT. Table 5 summarizes the results.

*Table 5.* The results of experimental comparison of network architectures in 9x9 Go. Columns represent training simulator evaluations and percentages denote win rates against the anchored baseline.

| Architecture | 200M | 400M | 600M | 800M |
| --- | --- | --- | --- | --- |
| (i) Shared-Backbone Single-Head | 3 % | 2 % | 1 % | 1 % |
| (ii) Separate-Backbones Separate-Heads (2x Computation Cost) | 61 % | 85 % | 87 % | 89 % |
| (Original KLENT) Shared-Backbone Separate-Heads | 53 % | 80 % | 85 % | 89 % |

First, variant (i) shows that fully sharing an output representation for policy and action-value is substantially less effective in our setting. This suggests that, although the two quantities are closely related, a single shared head can be too restrictive to effectively learn them. We also tested an EMA-based version of this design, but observed no improvement. These results indicate that separating the policy and action-value heads is beneficial here.

Second, for (ii), the comparison should be made in terms of computation cost. At the same number of simulator evaluations, variant (ii) shows a higher win rate in the early stages of training than the original shared-backbone separate-heads architecture, with both reaching the same win rate at 800M evaluations. However, variant (ii) requires about twice as much computation time per evaluation because it trains two independent networks. When comparing these performances at equivalent compute budgets, the original shared-backbone separate-heads architecture at 400M (80%) and 800M (89%) evaluations achieves higher win rates than variant (ii) at 200M (61%) and 400M (85%) evaluations, respectively. Therefore, we consider that the shared-backbone design is still more effective from the perspective of computational efficiency for training.

## I. Experimental Comparison with Other Regularized Policy Optimization Approaches

We compared KLENT with closely related regularized policy optimization alternatives in 9x9 Go. Apart from PPO (Schulman et al., 2017), which is already included in the main paper as a representative regularized policy optimization baseline, we further evaluated the following two approaches:

- Forward-KL: a variant of KLENT using forward-KL regularization.

- Entropy Reward: a variant of KLENT using entropy-only regularization, with entropy incorporated into return calculation.

*Table 6.* The results of experimental comparison with other regularized policy optimization approaches in 9x9 Go. Columns represent training simulator evaluations and percentages denote win rates against the anchored baseline.

| Method | 200M | 400M | 600M | 800M |
| --- | --- | --- | --- | --- |
| Forward-KL | 29% | 54% | 56% | 61% |
| Entropy Reward | 5% | 8% | 13% | 14% |
| KLENT | 53% | 80% | 85% | 89% |

The results in Table 6 show that KLENT consistently performs best among these closely related regularized policy optimization variants. We believe they support that the reverse-KL + entropy design in KLENT is not an arbitrary combination of known techniques, but an empirically effective design choice for efficient self-play learning in board games.

We also evaluated Discrete SAC (Christodoulou, 2019) and V-MPO (Song et al., 2020), which are discrete counterparts of SAC (Haarnoja et al., 2018) and MPO (Abdolmaleki et al., 2018), but learning remained unstable in 9x9 Go and neither exceeded a 1% win rate after 800M evaluations. While we do not claim these methods are universally ineffective, these results suggest that obtaining strong self-play performance with standard model-free regularized RL methods is non-trivial.

## J. Experimental Comparison with Recent Model-Free Approaches

We ran additional experiments with two recent model-free approaches: Munchausen DQN (Vieillard et al., 2020b) and GRPO (Shao et al., 2024), which are prominent in discrete action domains such as Atari and the language domain.

*Table 7.* The results of experimental comparison with recent model-free approaches in 9x9 Go. Columns represent training simulator evaluations and percentages denote win rates against the anchored baseline.

| Method | 200M | 400M | 600M | 800M |
|---|---|---|---|---|
| Munchausen DQN | 8 % | 5 % | 3 % | 2 % |
| GRPO | 1 % | 1 % | 2 % | 3 % |
| KLENT | 53 % | 80 % | 85 % | 89 % |

The results in Table 7 show that KLENT consistently achieves higher win rates than these recent model-free baselines.

## K. Additional Evaluation on the Reliability of the AlphaZero Implementation

### K.1. Reliability of the Pgx Implementation as a Baseline

In this study, we adopt the Pgx implementation as the baseline for AlphaZero-family methods. The original AlphaZero implementation by its authors is not publicly available. Similarly, for Gumbel AlphaZero, only the MCTS technique has been released through the `Mctx` library, and the full training pipeline is not open-sourced. Therefore, reproducing the full experimental setup of AlphaZero-family methods requires either relying on third-party open-source implementations or building one from scratch. To the best of our knowledge, Pgx is the only open-source implementation that satisfies all of the following criteria:

- **Peer-reviewed implementation**: Pgx was accepted to the NeurIPS 2023 benchmark track, indicating that its experimental setup has undergone peer review.

- **Evaluated across multiple environments**: Pgx has been tested on five different board games, not just a single domain. This suggests that the implementation is robust and not reliant on environment-specific tricks.

- **Performance comparison against other agents**: According to the Pgx paper, its baseline agent outperforms `pachi`, a reasonably strong Go engine.

- **Use of the `Mctx` library for MCTS**: Pgx utilizes the `Mctx` library for its MCTS technique, ensuring consistency with the Gumbel AlphaZero implementation, which was developed by some of the original AlphaZero authors.

For these reasons, we consider Pgx to be a reliable and robust open-source implementation of AlphaZero-family methods, and adopt it as the baseline in our experiments.

### K.2. Performance Comparison with Other Implementations

To strengthen the credibility of the AlphaZero and baseline implementations used in this study, we conducted a comparative evaluation against a well-known open-source implementation available at https://github.com/suragnair/alpha-zero-general. This repository provides pretrained models for several games, including $8 \times 8$ Othello. We used the provided checkpoint file

`pretrained_models/othello/8x8_100checkpoints_best.pth.tar` to construct an evaluation agent. We conducted a round-robin tournament involving the following four agents, where each pair played 100 games. Draws were counted as 0.5 wins for each agent.

- **Random**: An agent that selects legal moves uniformly at random.

- **AlphaZero-General**: An agent that follows the policy from the above checkpoint of `alpha-zero-general`.

- **Pgx Baseline**: The baseline agent used throughout our experiments.

- **Pgx's AlphaZero**: Our implementation of AlphaZero using the Pgx framework, trained with 800 million simulator evaluations.

The number of wins for each agent against the others is shown in Table 8. Each cell indicates the number of wins achieved by the row agent when playing against the column agent. As shown in the table, AlphaZero-General achieves a 97% win

*Table 8.* Win rates among AlphaZero implementations and baselines in Othello.

|  | Random | AlphaZero-General | Pgx Baseline | Pgx's AlphaZero |
|---|---|---|---|---|
| Random | – | 3 % | 0 % | 3 % |
| AlphaZero-General | 97 % | – | 17 % | 13 % |
| Pgx Baseline | 100 % | 83 % | – | 42 % |
| Pgx's AlphaZero | 97 % | 87 % | 58 % | – |

rate against the random agent, confirming that it is significantly stronger than random. However, both the Pgx Baseline and Pgx's AlphaZero implementation clearly outperform AlphaZero-General, achieving win rates of 83% and 87% respectively. These results support the reliability and strength of the implementations used in our experiments.

## L. Extended Experiments on Rollout Counts and Training Budgets for AlphaZero

This section presents additional experiments to examine how AlphaZero's performance is affected by the number of rollouts per move and the total training budget.

### L.1. Performance of AlphaZero with Varying Rollout Counts in 9x9 Go

AlphaZero performs Monte Carlo Tree Search (MCTS) at each move, where the number of rollouts corresponds to the number of simulator evaluations used per search. We investigated how this parameter affects learning efficiency.

The experiments were conducted in the 9x9 Go environment, using rollout counts of 2, 4, 8, 16, 32, and 64. The total number of simulator evaluations used during training was fixed at 200M, 400M, 600M, and 800M. Evaluation was performed by measuring the win rate against a fixed baseline agent. Note that for a fixed training budget, increasing the rollout count reduces the number of parameter updates, since each update consumes a number of simulator evaluations proportional to the rollout count. This highlights a trade-off: deeper search per move comes at the cost of fewer parameter updates. The results are shown in Table 9.

The results indicate that in 9x9 Go, setting the rollout count to around 8 leads to the most efficient learning for AlphaZero. Nevertheless, even when the rollout count is optimized, KLENT achieves substantially higher performance under the same training budget, highlighting its superior efficiency.

### L.2. Performance of AlphaZero with Varying Rollout Counts in 19x19 Go

We also tuned the number of rollouts in 19x19 Go with values of 4, 16, 64, and 256. As shown in Figure 14, 16 rollouts achieved the most efficient learning. Accordingly, we reported this result as the performance of AlphaZero in Figure 10 in Section 6.3.

*Table 9.* Performance of AlphaZero with different rollout counts (9x9 Go). Each entry shows the win rate (%) against the baseline agent.

| Simulator Evaluations | 200M | 400M | 600M | 800M |
|---|---|---|---|---|
| AZ (2 rollouts) | 7 % | 7 % | 7 % | 8 % |
| AZ (4 rollouts) | 16 % | 35 % | 51 % | 61 % |
| AZ (8 rollouts) | 20 % | 39 % | 56 % | 69 % |
| AZ (16 rollouts) | 15 % | 28 % | 42 % | 57 % |
| AZ (32 rollouts) | 6 % | 13 % | 20 % | 34 % |
| AZ (64 rollouts) | 5 % | 7 % | 11 % | 15 % |
| (cf: KLENT) | **53 %** | **80 %** | **85 %** | **89 %** |

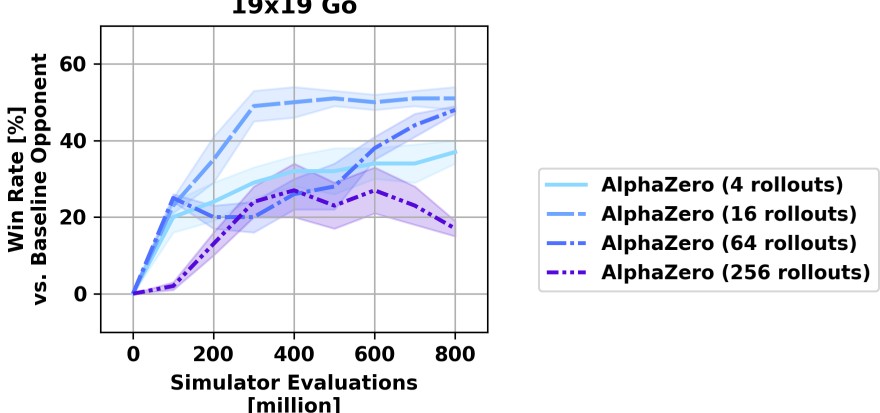

*Figure 14.* The results of rollout count tuning in 19x19 Go. 16 rollouts achieve the most efficient learning.

## L.3. Performance of AlphaZero with Increased Training Budgets

We also conducted additional experiments to examine AlphaZero's asymptotic performance by increasing the total training budget. The experimental settings were the same as above, and the number of simulator evaluations was extended up to 4,800M. The results are presented in Table 10.

*Table 10.* Performance of AlphaZero under increased training budgets (9x9 Go). Each entry shows the win rate (%) against the baseline agent.

| Simulator Evaluations | 800M | 1,600M | 2,400M | 3,200M | 4,000M | 4,800M |
|---|---|---|---|---|---|---|
| AZ (2 rollouts) | 8 % | 18 % | 17 % | 18 % | 19 % | 18 % |
| AZ (4 rollouts) | 61 % | 75 % | 75 % | 85 % | 86 % | 85 % |
| AZ (8 rollouts) | 69 % | 79 % | 85 % | 85 % | 85 % | 86 % |
| AZ (16 rollouts) | 57 % | 80 % | 83 % | 88 % | **89 %** | **89 %** |
| AZ (32 rollouts) | 34 % | 60 % | 71 % | 78 % | 83 % | 83 % |
| AZ (64 rollouts) | 15 % | 34 % | 51 % | 59 % | 67 % | 72 % |
| (cf: KLENT) | **89 %** | – | – | – | – | – |

These results show that AlphaZero reaches approximately 89% win rate when the total training budget is increased to around 3,200M to 4,000M simulator evaluations. This confirms the intuitive expectation that AlphaZero can achieve strong asymptotic performance given sufficient training budget. At the same time, KLENT achieves comparable performance using only 800 million simulator evaluations, which is approximately four to five times fewer than those required by AlphaZero, underscoring its efficiency advantage.

## M. Strength Scaling with Additional Test-time Computation

Additional simulations during test time can improve the strength of agents. In this section, we investigate how performance scales with the number of simulations for models trained with KLENT and those trained with Gumbel AlphaZero in 9x9 Go. For both methods, parameters trained with 800 million simulator evaluations are used. We adopt an off-the-shelf Gumbel AlphaZero Monte Carlo Tree Search (MCTS) for test-time computation, applying the same procedure to both sets of parameters. While Gumbel AlphaZero learns policy and state-value networks, KLENT trains policy and action-value networks. To address this difference, for KLENT, the inner product of the policy and action-value is used as the state-value estimate during MCTS. The anchored baseline opponent uses parameters provided by Pgx and runs with 800 simulations. Koyamada et al. (2023) have reported that this agent has achieved 62 wins and 38 losses against Pachi (Baudiš & Gailly, 2011) with 10,000 simulations. We measure the win rates of the evaluated target agents, using either KLENT or Gumbel AlphaZero parameters, under 0, 16, 32, 64, 100, 200, 400, and 800 simulations. Here, 0 indicates that the agent conducts no search and deterministically chooses action solely based on its policy network. In this experiment, the evaluation is conducted for 100 matches. The win rates are measured with three random seeds and the mean and the standard deviation are plotted.

The results are shown in Figure 15, where the horizontal axis represents the number of simulations and the vertical axis represents win rates against the anchored baseline. KLENT demonstrates that it can effectively scale its strength with test-time computation. In this experiment, we calculate the state-values from policy and action-value estimates as

$$V^\pi(s) = \sum_a \pi(a|s)Q^\pi(s,a),$$

which is based on the policy $\pi$ we actually use for rollout.

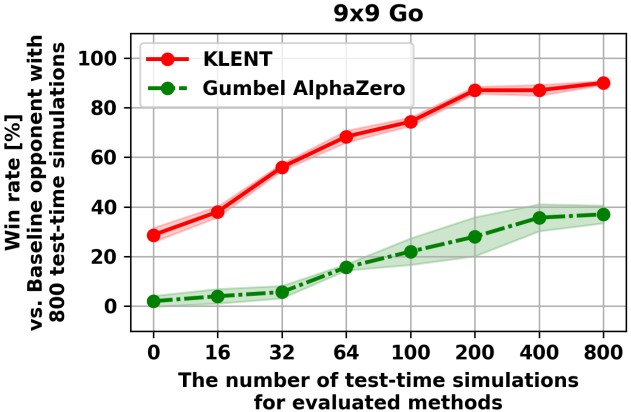

*Figure 15.* Performance changes with increased test-time computation budget. The simulation budget of the anchored baseline opponent is fixed at 800. The horizontal axis represents the simulation budget for the evaluated agents, while the vertical axis shows their win rate against the anchored opponent. The results demonstrate that agents using parameters trained with KLENT can scale their strength as the number of test-time simulations increases.

**Wall-clock Inference Time:** An evaluation with equal wall-clock search time is a fair condition for performance comparison. Actually, as we use the same ResNet architecture for each method, the wall-clock search time is proportional to the test-time rollout count. Therefore, the equal wall-clock search time comparison is equivalent to our equal rollout count comparison. To further verify this point, we have measured the wall-clock time spent for each action selection and summarized in the following table.

The results show there is no significant difference between wall-clock inference time of Gumbel AlphaZero and KLENT + test-time MCTS. Therefore, we conclude that the evaluation with fixed rollout counts is equivalent to equal wall-clock time evaluation.

*Table 11.* Test-time rollout counts and wall-clock inference time.

| Rollout Count | Gumbel AlphaZero | KLENT + Test-Time MCTS |
|---|---|---|
| 0 | $(7.128 \pm 0.400) \times 10^{-4}$ | $(7.157 \pm 0.084) \times 10^{-4}$ |
| 16 | $(2.601 \pm 0.013) \times 10^{-2}$ | $(2.603 \pm 0.012) \times 10^{-2}$ |
| 32 | $(5.403 \pm 0.019) \times 10^{-2}$ | $(5.359 \pm 0.026) \times 10^{-2}$ |
| 64 | $(1.093 \pm 0.006) \times 10^{-1}$ | $(1.100 \pm 0.005) \times 10^{-1}$ |
| 100 | $(1.732 \pm 0.007) \times 10^{-1}$ | $(1.734 \pm 0.007) \times 10^{-1}$ |
| 200 | $(3.435 \pm 0.010) \times 10^{-1}$ | $(3.444 \pm 0.017) \times 10^{-1}$ |
| 400 | $(6.918 \pm 0.041) \times 10^{-1}$ | $(6.970 \pm 0.021) \times 10^{-1}$ |
| 800 | $(1.391 \pm 0.009) \times 10^{0}$ | $(1.389 \pm 0.004) \times 10^{0}$ |

*Table 12.* The results of head-to-head matches against GnuGo and Pachi.

| Anchored Opponent | Winrate of KLENT's side |
|---|---|
| GnuGo (Level 10) | 100 % |
| Pachi (10K rollouts) | 81 % |

## N. Head-to-Head Matches

### N.1. Evaluation against Pachi and GnuGo in 9x9 Go

In the domain of 9x9 Go, we conducted additional head-to-head experiments against GnuGo and Pachi, which are baselines confirmed to have been used in prior studies. The detailed configurations of these agents are provided below.

- Evaluated Agent

  - KLENT: The model trained with KLENT. Similarly to Appendix M, Gumbel AlphaZero was employed as the search algorithm at test time, with the number of rollouts set to 2,000 (approximately two seconds per move). For the neural network parameters, we used the model trained by KLENT with 800M simulator evaluations. While the MCTS in Gumbel AlphaZero requires estimates of the policy and state value, KLENT's neural network estimates the policy and action values. To account for this difference, we used the inner product of the policy and action-value predictions as the state-value estimate.

- Anchored Opponent

  - GnuGo (Bump et al., 2005): A classical and lightweight MCTS-based Go engine. The strength level was set to 10 (the strongest level), following the evaluation setting in prior work (Hessel et al., 2021).
  - Pachi (Baudiš & Gailly, 2011): A fairly strong MCTS-based Go engine. This program has been reported to have the strength of a KGS 7-dan player in 9x9 Go (Baudiš & Gailly, 2018), which corresponds to the top 0.5–1% of players on Kiseido Go Server. The strength was set by configuring the MCTS rollout count to 10,000, consistent with the evaluation settings in prior work (Hessel et al., 2021; Danihelka et al., 2022; Koyamada et al., 2023).

Under these conditions, we conducted 100 games, and the win rate of KLENT is presented in Table 12. These results demonstrate the win rates against agents that have been used for evaluation in prior studies, and we believe they can serve as one of the credible reference points.

### N.2. Head-to-Head Match against AlphaZero in 19x19 Go

We additionally conducted direct head-to-head matches between the final checkpoints trained in Section 6.3. In this setting, the evaluation used MCTS with 800 rollouts per move. The AlphaZero checkpoint was trained with 16 rollouts, which was the strongest among the tested settings. Under this protocol, KLENT won all evaluation games, yielding a 100% win rate against AlphaZero trained with the same simulator budget. These result also support that KLENT can achieve efficient learning under a fixed training budget.

## O. Performance Comparison in Elo Ratings

While win rate was used as the primary metric for comparing trained agents in the main paper, for reference, we provide Elo scores in Figure 16. Specifically, we fix the Elo score of the Pgx baseline agent at $R_0 = 1000$, and apply the following standard formula for Elo rating:

$$R = 400 \log_{10} \left( \frac{W}{L} \right) + R_0,$$

where $W$ denotes the win rate against the Pgx baseline and $L = 1 - W$ is the corresponding loss rate. Since the mapping from win rate to Elo is monotonic, this transformation does not alter our primary claim that KLENT outperforms the baselines under a fixed computational budget. However, Elo scores must be interpreted with care, as they are highly sensitive to the composition of the tournament pool. Indeed, in our preliminary experiments, we observed that Elo ratings of fixed agents could vary significantly when the set of evaluated agents is modified. This sensitivity has also been pointed out in prior works (Balduzzi et al., 2018; Liu et al., 2025; Lanctot et al., 2025). These studies highlight that Elo ratings can be manipulated by adding redundant or biased agents, even when anchor points are fixed. Therefore, cross-paper comparisons of Elo scores require identical tournament configurations, which is difficult in our case since neither the full tournament details of the Pgx implementation nor those of Gumbel AlphaZero are publicly available. For this reason, we present Elo scores only as supplementary information.

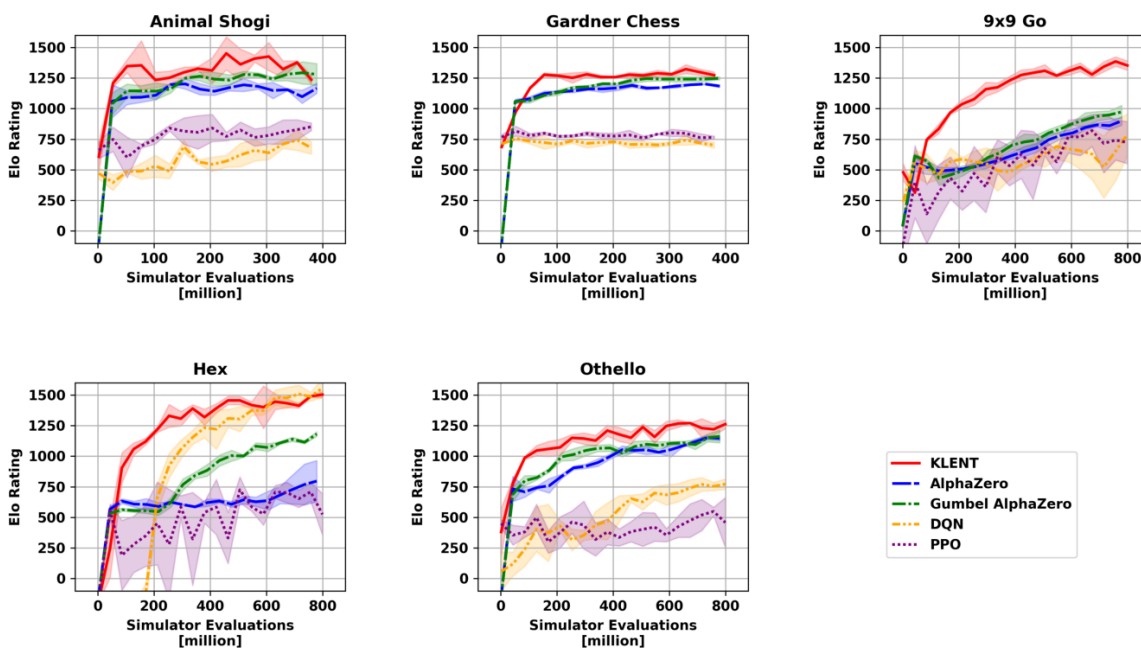

*Figure 16.* **Performance comparison in Elo scores.** Win rates are converted by fixing the Pgx baseline to Elo 1000. Note that Elo-based cross-paper comparisons are unreliable due to sensitivity to tournament configurations.

## P. Computational Requirements

For overall experiments, we have spent approximately 2,000 GPU hours on NVIDIA A100 GPU in total to run the main experiments in Figure 6 to 10. KLENT algorithm can be run on a single NVIDIA A100 GPU. This section describes the computational and memory requirements of the algorithm.

**Memory Usage**  KLENT stores improved policies in a replay buffer for reuse. In our experiments, memory usage was not an issue on a single A100 GPU with 80 GB of memory. Even when memory becomes a limiting factor, this issue can be

mitigated using a sparse representation. Since the improved policy assigns non-zero probabilities only to legal actions and sets all others to zero, sparse storage formats can significantly reduce memory consumption.

To illustrate this, we collected states from 10,000 games played by baseline agents implemented with Pgx and computed the average and maximum number of legal actions per game. The results are shown in Table 13.

*Table 13.* Statistics of legal actions collected from 10,000 games for each environment.

| Game | Action Space Size | Mean Legal Actions | Max Legal Actions |
|------|------|------|------|
| Animal Shogi | 132 | 7.5 | 36 |
| Gardner Chess | 1,225 | 9.5 | 40 |
| 9x9 Go | 82 | 42.3 | 82 |
| Hex | 122 | 90.6 | 121 |
| Othello | 65 | 8.0 | 22 |

These results indicate that the number of legal actions is often much smaller than the full action space. Therefore, sparse representations provide an effective solution in memory-constrained settings.

**Computation Time**    One of KLENT's strengths lies in its training efficiency. For example, in the 9x9 Go environment, KLENT reduced the time required to surpass the baseline agent by more than 25% compared to Gumbel AlphaZero and AlphaZero.

This efficiency stems from KLENT requiring fewer simulator interactions and neural network evaluations per training sample. As a result, it offers practical advantages in terms of wall-clock training time and computational cost.

# Part IV: Further Discussions

## Q. Extension to Stochastic Environments

In our experiments, we focused on the five board games in Table 1, all of which have deterministic state transitions. However, the formulation of KLENT assumes only a finite-action MDP and does not require deterministic transitions. Therefore, we consider that KLENT can be naturally applied to stochastic environments, and such applications are an important future research direction.

## R. Extension to Continuous Action Spaces

In finite-action settings, the updated policy is obtained by explicitly normalizing the right-hand side of Equation 3. However, in continuous action spaces, this normalization becomes intractable due to the integral required for the partition function, and an alternative approach is needed. A natural extension is a sampling-based update, which weights actions sampled from the current policy $\pi_\theta$ based on $Q_\theta$ and $\pi_\theta$ to approximate the distribution. This idea conceptually aligns with MPO (Abdolmaleki et al., 2018), which carries out policy improvement through weighted empirical distributions over sampled actions. We therefore consider that a sampling-based approximation is a feasible candidate for extending KLENT to continuous action spaces.

