# OpenReview forum: "Revisiting Regularized Policy Optimization for Stable and Efficient Reinforcement Learning in Two-Player Games"
_ICML.cc/2026/Conference — ICML 2026 regular_

### Official Review · Reviewer_eFut · 2026-03-03

**Soundness:** 3
**Presentation:** 3
**Significance:** 3
**Originality:** 3
**Overall Recommendation:** 5
**Confidence:** 4

**Summary:**

This paper proposes the KLENT algorithm, which integrates reverse Kullback-Leibler (KL) regularization with an entropy term in the policy update rule. The authors provide novel theoretical convergence guarantees for two-player zero-sum games in both normal-form and finite-length sequential settings, improving upon prior results (e.g., Sokota et al., 2022) by covering a broader range of regularization coefficients. Empirical evaluations on five medium-scale board games demonstrate that KLENT achieves more efficient learning compared to existing search-based and model-free baselines.

**Compliance With Llm Reviewing Policy:**

Affirmed.

**Final Justification:**

I keep my original positive evaluation on this paper.

**Key Questions For Authors:**

I have outlined my main concerns in the Strengths and Weaknesses sections above. Here are a few additional questions for the authors:
1. The theoretical convergence guarantees (Theorems 5.1 and 5.2) are derived specifically for two-player zero-sum games. Could these results be extended to more general settings, such as two-player general-sum games or multi-player games? If not, what are the key technical obstacles (e.g., loss of antisymmetry in the payoff structure or increased complexity in multi-agent dynamics)?
2. The empirical results in Section 6.1 report average performance across the five board games in Figure 5, with more detailed per-game learning curves provided in Figure 6. However, the textual discussion of these per-game results remains quite brief (e.g., only short mentions for Animal Shogi and Gardner Chess). Could the authors expand the analysis in the main text to provide more granular insights into KLENT’s relative strengths and weaknesses across individual games? Specifically, what game characteristics (such as action space size, branching factor, typical episode length, degree of strategic depth, or sensitivity to search depth) seem to favor KLENT over search-based baselines, and why? This qualitative discussion would help readers better understand the scope and applicability of the method, even without additional experimental results.
3. In the ablation study (Section 6.2), the ENT Only baseline (entropy regularization without reverse KL) is shown to underperform KLENT. To further substantiate the importance of gradual policy updates via reverse KL, could the authors include a visualization of the average KL divergence (e.g., D_KL(π′ || π) or similar) between consecutive policy updates for the ENT Only variant, compared to KLENT? Such a plot would provide direct evidence of policy instability in the absence of reverse KL regularization.

**Limitations:**

yes

**Strengths And Weaknesses:**

* Soundness: Good, but not sufficient.
The experimental results highlight two key advantages of KLENT: (1) convergence under relatively mild assumptions on regularization coefficients, and (2) improved sample efficiency compared to baselines. The first advantage is well supported by Theorem 5.1, which advances the theoretical convergence boundary beyond prior work (e.g., Sokota et al., 2022) by covering a broader range of (α, β) pairs. However, the second advantage—superior sample efficiency—lacks corresponding theoretical support in the form of convergence rate analysis or sample complexity bounds. Additionally, most baseline algorithms (e.g., DQN, PPO, AlphaZero) are relatively dated; comparisons against more recent model-free or search-efficient methods could strengthen the empirical claims.

* Presentation: Good.
The overall structure of the paper is clear and logical. I have only a few minor suggestions for improvement:
1. The formulation and theoretical analysis implicitly rely on the assumption that the state space forms a DAG (no cycles, unique depth-to-termination per state), which is valid for board games but is never explicitly stated. Adding a brief remark in Section 2 or 5.2 would improve clarity and rigor.
2. The theoretical results are specifically for two-player zero-sum (2p0s) games and their extensions. The current claims of providing results for “Normal-Form Games” and “Finite-Length Games” are slightly imprecise, as these terms are broader; clarifying that the analysis targets 2p0s instances of these settings would be more accurate.
3. The theoretical analysis proves convergence to a fixed point but does not fully characterize the properties of that fixed point (e.g., its relationship to the Nash equilibrium of the original game, approximation quality relative to the unregularized optimum, etc.). While referencing existing results on entropy-regularized equilibria is acceptable, a short discussion of these aspects would make the theoretical contribution more complete.

* Significance: Good.
Advancing the theoretical convergence boundary is a meaningful contribution in itself. However, the significance is somewhat limited by the fact that KLENT’s empirical advantage over strong search-based baselines (particularly AlphaZero-style methods) remains modest in terms of final performance, even though training is clearly more efficient in the medium regime.

* Originality: Good.
The combination of reverse KL + entropy regularization is not entirely novel, but the improved convergence guarantee and the demonstration of strong model-free performance without any search during training represent a solid and original step forward in this line of work.

---

> ### Author Rebuttal · Authors · 2026-03-29
>
> We thank the reviewer for the careful and thoughtful review. We respond to the suggestions and questions below.
>
> **Suggestions on Soundness of the Paper**
>
> Regarding the convergence rate, since the proof of Theorem 5.1 is based on a Jacobian-based contraction argument around the fixed point, the theorem in fact implies a linear convergence rate under the same condition. We will state this convergence rate explicitly in Theorem 5.1 and Appendix B in the revised version.
>
> We thank the reviewer for the baseline suggestions. For search-based methods, we included TRPO AlphaZero [1] and Gumbel AlphaZero [2] as recent search-efficient approaches. For model-free methods, we ran additional experiments with Munchausen DQN [3] and GRPO [4], which are prominent in discrete action domains such as Atari and the language domain. The table below presents 9x9 Go win rates against the anchored baseline at each training simulator evaluation budget.
>
> |Method|200M|400M|600M|800M|
> |-|-:|-:|-:|-:|
> |Munchausen DQN|8 %|5 %|3 %|2 %|
> |GRPO|1 %|1 %|2 %|3 %|
> |KLENT|53 %|80 %|85 %|89 %|
>
> The results show that KLENT consistently achieves higher win rates than these recent model-free baselines. These additional results strengthen our experimental comparison, and we will include them in the revised version.
>
> **Suggestions on Presentation of the Paper**
>
> As the reviewer kindly pointed out, Sections 5.2 and B.2 assume a directed acyclic graph (DAG) state transition structure, and Section 5 and Appendix B assume two-player zero-sum games. Regarding the characteristics of the fixed point, our update rule's fixed point in Equation 6 is an entropy-regularized Nash equilibrium [5]. We agree that it is worth mentioning that this regularized equilibrium converges to the original game's Nash equilibrium when the regularization coefficient $\alpha$ goes to $0$.
>
> Following the kind suggestion, we will include these clarifications and discussions in the revised version.
>
> **Q1: On Extensions to General-Sum and Multi-Player Settings**
>
> Theorems 5.1 and 5.2 rely on different structures, so we address them separately.
>
> Theorem 5.1 extends to general-sum settings. Our proof uses the two-player zero-sum structure, where player 1's payoff matrix $A$ defines player 2's payoff as $-A^\top$. This can be adapted to general-sum games by replacing $-A^\top$ with a distinct payoff matrix $A'$ for player 2. While the final convergence condition will depend on both $A$ and $A'$, the analysis of the Jacobian's spectral radius around the fixed point remains valid.
>
> Extending Theorem 5.1 to multi-player games is challenging. Two-player games use a two-dimensional payoff matrix, whereas three or more players require a higher-order payoff tensor. The contraction mapping approach may remain effective, but analyzing the higher-order mapping requires extending the core of our proof beyond Jacobian analysis.
>
> Theorem 5.2 depends mostly on a bounded horizon and backward induction, relying less on the two-player zero-sum structure. We consider that it could naturally extend to multi-player general-sum settings.
>
> We will add the discussion above in the revision.
>
> **Q2: Qualitative Analysis of the Per-Game Learning Curves**
>
> The results in Figure 6 show that KLENT is relatively stronger in games with a larger branching factor. KLENT and search-based methods are competitive in Animal Shogi and Gardner Chess (mean legal actions 7.5 and 9.5). Conversely, KLENT has an advantage in 9x9 Go and Hex (where the means are 42.3 and 90.6). This aligns with the intuition that MCTS-based methods spend more simulator evaluation budget when the branching factor is large, whereas KLENT avoids the look-ahead search. We will revise Section 6.1 to include this discussion.
>
> **Q3: Evolution of KL Divergence in KLENT and the ENT-Only Variant**
>
> Following the suggestion, we examined this in 9x9 Go, where the performance gap between KLENT and ENT Only is large. The table below shows the evolution of the average KL divergence $D_{\mathrm{KL}}(\pi' || \pi)$, confirming it is consistently larger for ENT Only.
>
> |Method|200M|400M|600M|800M|
> |-|-:|-:|-:|-:|
> |KLENT|0.0063|0.0035|0.0033|0.0030|
> |ENT Only|0.1143|0.0618|0.0436|0.0548|
>
> These results support the reviewer's point that policy updates are less gradual without reverse KL regularization. In the revision, we will visualize this quantity with a plot similar to Figure 8 as part of the ablation study in Section 6.2.
>
> **References**
>
> [1] : Grill et al. "Monte-Carlo Tree Search as Regularized Policy Optimization." ICML 2020.
>
> [2] : Danihelka et al. "Policy improvement by planning with Gumbel." ICLR 2022.
>
> [3] : Vieillard et al. "Munchausen Reinforcement Learning." NeurIPS 2020.
>
> [4] : Shao et al. "DeepSeekMath: Pushing the Limits of Mathematical Reasoning in Open Language Models." arXiv 2024.
>
> [5] : McKelvey and Palfrey. "Quantal response equilibria for normal form games." 1995.
>
> ---
> We thank the reviewer again for the detailed and comprehensive review.

---

> > ### Author Rebuttal · Reviewer_eFut · 2026-04-03
> >
> > I thank the authors' response and remain my positive score on this paper.

---

> > > ### Author Response · Authors · 2026-04-04
> > >
> > > We are pleased that we were able to address the reviewer’s concerns and that they have kindly maintained their positive score. We believe these additional discussions and experiments will further strengthen the paper, and we will include them in the revised version. We sincerely thank the reviewer once again for their detailed and highly constructive feedback.

---

### Official Review · Reviewer_5Ag7 · 2026-03-10

**Soundness:** 3
**Presentation:** 3
**Significance:** 3
**Originality:** 2
**Overall Recommendation:** 4
**Confidence:** 4

**Summary:**

This paper proposes a model-free reinforcement learning algorithm called KLENT, which is based on regularized policy optimization with reverse KL divergence and entropy regularization. The goal is to achieve stable and competitive policy learning using significantly fewer training resources than search-based algorithms. Reverse KL divergence is used for gradual policy updates, while entropy regularization is used for sustained exploration. From a theoretical perspective, the authors prove the stability and convergence of KLENT's policy update rule in normal-form games and finite-length games. From an empirical perspective, the authors validate that the proposed algorithm achieves higher learning efficiency across multiple board games.

**Compliance With Llm Reviewing Policy:**

Affirmed.

**Final Justification:**

Since the author addressed my concerns, I am happy to increase my rating from 3 to 4.

**Key Questions For Authors:**

1. Why did the authors choose the combination of reverse KL regularization and entropy regularization instead of other combinations?

2. Is it possible to add experimental comparisons with other regularized policy optimization methods?

**Limitations:**

yes

**Strengths And Weaknesses:**

Strengths:

1. Applying a model-free reinforcement learning algorithm based on reverse KL divergence and entropy regularized policy optimization to solve two-player zero-sum games, especially in the domain of board games, is a novel application attempt.

2. The authors provide novel theoretical convergence analyses for this specific combination.

3. The experiments demonstrate that this combination possesses significant advantages in board games.

Weaknesses:

1. My main concern lies in the technical novelty. As stated by the authors in Section 3.1, there are already many methods and theoretical analyses regarding the combination of reverse KL regularization and entropy regularization. Furthermore, $\lambda$-returns are also a mature technique. The authors need to provide an in-depth explanation of why combining these techniques is particularly effective in board games, rather than it being just a simple combination.

2. The experiments lack comparisons with other model-free reinforcement learning algorithms based on regularized policy optimization. Reverse KL divergence and entropy regularization are merely two techniques within regularized policy optimization.

---

> ### Author Rebuttal · Authors · 2026-03-28
>
> We thank the reviewer for the careful review and constructive feedback. We respond to the main points below.
>
> **W1: Scope of Our Novelty Claim**
>
> We would like to clarify that our main contributions lie not in proposing entirely new algorithmic ingredients, but rather in theoretically proving novel convergence guarantees and empirically demonstrating that a model-free approach can achieve efficient learning in the board game domain, as the reviewer kindly mentioned in the Summary and Strengths. Regarding the question on the reasons for the specific combination, we elaborate on them in our response to Q1 below.
>
> **Q1: Motivation for Combining Reverse-KL and Entropy Regularization**
>
> We have chosen to combine reverse KL and entropy regularization for the following three reasons.
>
> First, both regularizations play important roles in the self-play learning setting. In self-play, optimizing a policy against constantly changing opponents is a non-stationary problem requiring gradual updates to prevent abrupt policy changes. Furthermore, addressing the train-test distribution shift from unseen test-time opponents requires moderate exploration to prevent over-fitting to the policy of the agent itself. Reverse KL and entropy regularizations can address these non-stationarity and distribution shift, respectively.
>
> Second, this combination yields an explicit, closed-form analytical solution. While forward KL alone provides an analytical solution [1], it lacks explicit exploration. Conversely, combining forward KL and entropy requires the Lambert W function, preventing a closed-form expression without special functions [2]. In contrast, combining reverse KL and entropy provides a computable analytical solution, as shown in Equation 3.
>
> Third, this analytical solution yields favorable theoretical and empirical results. It enables the theoretical convergence guarantees in Section 5. Additionally, our main experiments show this combination consistently achieves efficient learning. The ablation study also confirms efficiency decreases if either component is missing, indicating the importance of using them together. These results further support our design choice.
>
> These are the reasons why we have chosen the combination of reverse KL and entropy regularization. To clarify these points for future readers, we will include these discussions in the revised version.
>
> **W2, Q2: Experimental Comparison with Other Regularized Policy Optimization Approaches**
>
> We agree that such comparisons are important. To directly address the reviewer’s concern, we compared KLENT with closely related regularized policy optimization alternatives in 9x9 Go. Apart from PPO, which is already included in the main paper as a representative regularized policy optimization baseline, we further evaluated the following two approaches:
>
> (i) Forward-KL: a variant of KLENT using forward-KL regularization.
>
> (ii) Soft Q-Learning: a variant of KLENT using entropy-only regularization, with entropy incorporated into return calculation.
>
> The table below summarizes the results. Columns represent training simulator evaluations and  percentages denote win rates against the anchored baseline.
>
> |**Method**|**200M**|**400M**|**600M**|**800M**|
> |-|-:|-:|-:|-:|
> |(i) Forward-KL|29 %|54 %|56 %|61 %|
> |(ii) Soft Q-Learning|5 %|8 %|13 %|14 %|
> |KLENT|53 %|80 %|85 %|89 %|
>
> These results show that KLENT consistently performs best among these closely related regularized policy optimization variants. We believe they directly support that the reverse-KL + entropy design in KLENT is not an arbitrary combination of known techniques, but an empirically effective design choice for efficient self-play learning in board games.
>
> We also evaluated Discrete SAC [3] and V-MPO [4], which are discrete counterparts of SAC [5] and MPO [6], but learning remained unstable in 9x9 Go and neither exceeded a 1% win rate after 800M evaluations.  While we do not claim these methods are universally ineffective, these results suggest that obtaining strong self-play performance with standard model-free regularized RL methods is non-trivial.
>
> These additional experiments strengthen our evaluation by comparing KLENT with other regularized policy optimization methods. We will include these results in the revised version.
>
> **References**
>
> [1] : Grill et al. "Monte-Carlo Tree Search as Regularized Policy Optimization." ICML 2020.
>
> [2] : Chow. "What is a closed-form number?", 1999.
>
> [3] : Christodoulou. "Soft Actor-Critic for Discrete Action Settings." arXiv 2019.
>
> [4] : Song et al. "V-MPO: On-Policy Maximum a Posteriori Policy Optimization for Discrete and Continuous Control." ICLR 2020.
>
> [5] : Haarnoja et al. "Soft Actor-Critic: Off-Policy Maximum Entropy Deep Reinforcement Learning with a Stochastic Actor." ICML 2018.
>
> [6] : Abdolmaleki et al. "Maximum a Posteriori Policy Optimisation." ICLR 2018.
>
> ---
>
> We thank the reviewer again for the thoughtful and constructive feedback.

---

> > ### Author Rebuttal · Reviewer_5Ag7 · 2026-04-02
> >
> > Since the author addressed my concerns, I am happy to increase my rating from 3 to 4.

---

> > > ### Author Response · Authors · 2026-04-02
> > >
> > > We are happy to hear that we could resolve the reviewer’s concerns and that the reviewer has kindly updated their rating to a positive one. We believe that these additional discussions and experiments will strengthen the paper, and we will include them in the revised version. We sincerely thank the reviewer again for their detailed and highly constructive feedback.

---

### Official Review · Reviewer_o2Zz · 2026-03-13

**Soundness:** 4
**Presentation:** 3
**Significance:** 3
**Originality:** 3
**Overall Recommendation:** 5
**Confidence:** 4

**Summary:**

I really enjoyed reading this paper. They study two player games with regularized policy gradient methods (theoretical+empirical perspectives). They provide theoretical results for normal form and finite length games and also provide a whole suite of experimental evidence on board games with a practical regularized policy optimization (reverse KL) algorithm version which is pure model-free RL algorithm that achieves stable and competitive learning with significantly fewer training resources than search-based counterparts.

**Compliance With Llm Reviewing Policy:**

Affirmed.

**Final Justification:**

I updated my confidence because I feel that the rebuttal acknowledged my primary questions. The paper manages to provide some nice theory related to local convergence guarantees alongside strong experimental evidence in a variety of settings to show efficiency against search-based method counterparts. Therefore, I found the paper very sound with good presentation (especially after discussing part of the reasoning/intuition for where such efficiency gains stem from).

**Key Questions For Authors:**

1) Why do you think KLENT's model free approach is able to do things more effectively by directly learning the action value function and policy via neural networks using the regularized policy gradient approach? I see that the nice closed form solution for the policy based on the optimization objective relies on having good action value estimates--is it primarily benefitting from G_t^lambda estimation being simpler than MCTS?
2) How would you naturally extend this to continuous action spaces or to infinite horizon games (though I know most games may not require this)? The proof currently relies on being able to make use of the existence of a T_max for max steps to a terminal state--could one make use of a randomized T_max and connect that to some discounted infinite horizon settings?

**Limitations:**

yes

**Strengths And Weaknesses:**

Strengths
This paper provides a combination of strong experimental evidence and theoretical analysis. They analyze regularized policy gradient with reverse KL divergence and provide a natural update rule and algorithm based upon this. They provide strong experimental evidence of efficiency with this algorithm (which is game agnostic) compared to search-based counterparts, which is really exciting. They also provide theory related to convergence guarantees for both normal form and finite gains. As stated in the paper: three key techniques include: KL regularization for gradual policy updates, entropy regularization for exploration, and λ-returns for efficient and stable value function learning. They also provide very nice diagrams showcasing the properties of the system in parameter space

Weaknesses
As mentioned in the paper, the primary focus is efficiency as opposed to asymptotic performance with large compute where search based methods have shown quite impressive performance. Moreover, they seem to rely on a finite action space which limits scope for continuous action games. Theoretically, for normal form games a local convergence argument is primarily provided though it would be interesting to provide some type of analysis or comparison with something like convergence rates between search-based methods (even under some set of assumptions) and KLENT. Also, in the proof of Theorem B.10, it would be nice to formalize this argument a bit more for the reader even though I agree with the statement "Therefore, by repeatedly applying the policy update rule, the policy is updated into the soft
(Boltzmann) form, and as the number of updates n → ∞, it converges to the desired Boltzmann policy."

---

> ### Author Rebuttal · Authors · 2026-03-28
>
> We thank the reviewer for their encouraging feedback and are honored that the paper was found enjoyable to read. Below, we address the questions and concerns raised in the review.
>
> **W1: Scope of Our Empirical Focus**
>
> As the reviewer kindly pointed out, this study does not preclude the effectiveness of powerful search-based approaches including AlphaZero, when massive computational resources such as thousands of GPUs are available. We consider that clarifying the scope of our empirical focus is important and therefore we have explicitly stated this in Section 7 to ensure our primary contributions are clearly conveyed.
>
> **W2: Extension to Continuous Action Spaces**
>
> We also consider the extension to continuous action spaces an interesting direction, and we address this in our response to Q2-1.
>
> **W3: Convergence Rate Analysis**
>
> We agree that making the convergence rate explicit would improve the presentation. Since the proof of Theorem 5.1 is based on a Jacobian-based contraction argument around the fixed point, the theorem in fact implies local linear convergence under the same condition. We will state this rate explicitly in Theorem 5.1 and Appendix B in the revised version. Although a direct comparison of convergence rates with search-based methods is not straightforward, as normal-form games are not typically used to analyze them, we believe this clarification is helpful for readers to theoretically understand the efficiency of KLENT.
>
> **W4: Strengthening the Proof of Theorem B.10**
>
> We acknowledge that the part of the proof of Theorem B.10 quoted by the reviewer can be made more formal. In this step, the sequence of policy logits induced by our policy update rule satisfies a linear recurrence with an explicit solution, from which the convergence to the Boltzmann policy follows. We will clarify this point accordingly in the revised version.
>
> **Q1: The Advantage of KLENT Approach**
>
> As the reviewer has kindly pointed out, the analytical policy update in Equation 3 relies on the quality of the action-value estimates. It is also true that estimating the $\lambda$-return $G_t^\lambda$ is computationally simpler than performing MCTS. We consider that the efficiency of our approach stems from the following points:
>
> * Regarding the policy update process, KLENT adopts an analytical update rule instead of using the results of MCTS. This reduces the number of simulator evaluations and neural network inferences required per action and improves the sample efficiency.
> * Regarding the optimization of the value learning target, reliability in value estimation is a challenge in model-free learning as the reviewer noted. To address this, KLENT uses the $\lambda$-return $G_t^\lambda$ as a target instead of the Monte Carlo returns used in existing search-based methods. Our ablation study also shows the effectiveness of $\lambda$-returns.
>
> We consider the design that combines analytical policy updates via regularization and stable value learning via $\lambda$-returns to be the main factor for achieving high learning efficiency.
>
> **Q2-1: Extension to Continuous Action Spaces**
>
> In finite-action settings, the updated policy is obtained by explicitly normalizing the right-hand side of Equation 3. However, in continuous action spaces, this normalization becomes intractable due to the integral required for the partition function, and an alternative approach is needed. A natural extension is a sampling-based update, which weights actions sampled from the current policy $\pi_\theta$ based on $Q_\theta$ and $\log \pi_\theta$ to approximate the distribution. This idea conceptually aligns with MPO [1], which carries out policy improvement through weighted empirical distributions over sampled actions. We therefore consider that a sampling-based approximation is a feasible candidate for extending KLENT to continuous action spaces.
>
> **Q2-2: Extension to Infinite-Horizon Games**
>
> We thank the reviewer for this insightful suggestion. Our current proof for finite-horizon games is based on backward induction and relies on the existence of $T_{\rm max}$, so it does not directly extend to infinite-horizon games. Therefore, a theoretical extension to infinite-horizon games would likely require a different analytical framework, for example one incorporating discounting or a stochastic horizon as suggested by the reviewer.
>
> We will include these discussions in the revised version.
>
> **References**
>
> [1] : Abdolmaleki, Abbas, et al. "Maximum a Posteriori Policy Optimisation." (ICLR 2018)
>
> ---
>
> We sincerely thank the reviewer again for their warm, encouraging words and highly constructive feedback. We hope the discussions above address the concerns and questions the reviewer has kindly raised.

---

> > ### Author Rebuttal · Reviewer_o2Zz · 2026-04-01
> >
> > Thanks for answering my questions--I updated my confidence based on the above!

---

> > > ### Author Response · Authors · 2026-04-02
> > >
> > > We are happy to hear that we could resolve the reviewer’s concerns and that the reviewer has kindly increased the confidence score while maintaining the positive rating. We believe that the discussions above will strengthen the paper, and we will include them in the revised version. We sincerely thank the reviewer again for their encouraging and highly constructive feedback.

---

### Official Review · Reviewer_CtYp · 2026-03-13

**Soundness:** 3
**Presentation:** 3
**Significance:** 3
**Originality:** 3
**Overall Recommendation:** 4
**Confidence:** 3

**Summary:**

The paper revisits a regularized RL method for two-player game. The authors theoretically prove the convergence guarantee of this method, and empirically evaluate its performance on five board games, demonstrating a comparable performance with less computation concumption compared to search-based methods.

**Compliance With Llm Reviewing Policy:**

Affirmed.

**Final Justification:**

The authors addressed my concerns regarding the network architecture through additional experiments.

**Key Questions For Authors:**

1. The current design uses a common NN but two seperate heads for the policy and the state-action values. If the action values are assumed static, the KL term does not change the convergence point of the policy, which is the same as the action value up to a scale $\alpha$ (as we use softmax parameterization for the policy), hence the policy can be viewed as a slowly updated version of action values. So my question is, how is the current method compared to (i) one single NN and one head for both the policy and the action values (possibly with EMA) and (ii) two seperate NNs for the policy and the action values, respectively?
2. Can KLENT be applied to more general and chanllenging tasks than two-player zero-sum game?

**Limitations:**

yes

**Strengths And Weaknesses:**

Strengths:
1. The paper studies an interesting problem.
2. The method is theoretically and empirically solid.

Weaknesses:
1. The proposed method lack novelty.
2. It requires more evidence to demonstrate the advantage of model-based RL methods over search-based methods.

---

> ### Author Rebuttal · Authors · 2026-03-28
>
> We thank the reviewer for the thoughtful review and constructive feedback. We also appreciate the reviewer’s recognition that the paper studies an interesting problem and is theoretically and empirically solid. We address the concerns and questions below.
>
> **W1: Scope of Our Novelty Claim**
>
> We would like to clarify that the novelty of our work lies not in the algorithmic ingredients themselves, but in the new theoretical and empirical characterization of this specific regularized update in two-player games. Theoretically, we establish a novel convergence guarantee in normal-form games that covers a broader range of regularization coefficients than prior work, matching the numerical results. Empirically, we show that in board games, where search-based training has been dominant, a pure model-free approach without search during training can still achieve high training efficiency, including a fourfold efficiency gain in our experiments. We will further clarify this scope of contribution in the revised version.
>
> **W2: Scope of the Empirical Claim**
>
> We respectfully clarify that we are not claiming the general superiority of model-free methods over search-based ones. Rather, our empirical contribution is to highlight the potential of model-free approaches through extensive experiments in the board-game domain, where look-ahead search has long been believed to be essential.
>
> **Q1: Architecture Choice**
>
> We thank the reviewer for this insightful question. In the main paper, we used the AlphaZero-style shared-backbone separate-head architecture throughout, in order to compare RL algorithms under a common setup.
>
> At the same time, we agree that the reviewer’s question is important, and we thus conducted additional experiments on 9x9 Go. To make the comparison precise, we interpret suggestion (i) as a single-network single-head variant where policy and action-value fully share one output representation, with both training losses applied to that shared head. We interpret (ii) as a fully separated two-network variant with independent backbones and heads for policy and action-value, so that no internal representation is shared. All other configurations were kept identical to those of the original KLENT. The table below summarizes the results. Columns show training simulator evaluations, and percentages denote win rates against the anchored baseline.
>
> |Architecture|200M|400M|600M|800M|
> |-|-:|-:|-:|-:|
> |(i) Shared-Backbone Single-Head|3 %|2 %|1 %|1 %|
> |(ii) Separate Backbones Separate Heads (2x Computation Cost)|61 %|85 %|87 %|89 %|
> |(Original KLENT) Shared-Backbone Separate-Heads|53 %|80 %|85 %|89 %|
>
> First, variant (i) shows that fully sharing an output representation for policy and action-value is substantially less effective in our setting. This suggests that, although the two quantities are closely related, a single shared head can be too restrictive to effectively learn them. We also tested an EMA-based version of this design, but observed no improvement. These results indicate that separating the policy and action-value heads is beneficial here.
>
> Second, for (ii), the comparison should be made in terms of computation cost. At the same number of simulator evaluations, variant (ii) shows a higher win rate in the early stages of training than the original architecture, with both reaching the same win rate at 800M evaluations. However, variant (ii) requires about twice as much computation time per evaluation because it trains two independent networks. When comparing these performances at equivalent compute budgets, the original architecture at 400M (80%) and 800M (89%) evaluations achieves higher win rates than variant (ii) at 200M (61%) and 400M (85%) evaluations, respectively. Therefore, we consider that the shared-backbone design is still more effective from the perspective of computational efficiency for training.
>
> These results provide further insights into suitable architectures for KLENT, and we will include them in the revised version.
>
> **Q2: Applicability Beyond Two-Player Zero-Sum Games**
>
> We also appreciate this question. We believe that KLENT can be applied to more general and challenging tasks than two-player zero-sum games, as its formulation only assumes a finite-action MDP. Potential tasks include other finite-action MDPs such as discrete optimization, algorithmic discovery, and mathematical proving, as we briefly mentioned in our introduction. As a side note regarding the theoretical extension, we discuss this in more detail in our response to Q1 from Reviewer eFut. We consider applying KLENT to these challenging tasks to be an important future direction, and we will include these discussions in the revised version.
>
> ---
>
> We thank the reviewer again for the helpful feedback.

---

> > ### Author Rebuttal · Reviewer_CtYp · 2026-04-03
> >
> > Thank you for the rebuttal. I have raised my scores.

---

> > > ### Author Response · Authors · 2026-04-04
> > >
> > > We are pleased that we were able to address the reviewer’s concerns and that they have kindly raised their score. We believe the above discussions and experiments will strengthen the paper, and we will include them in the revised version. We sincerely thank the reviewer once again for their valuable feedback.

---

### Decision · Program_Chairs · 2026-04-30

**Decision:**

Accept (regular)

**Comment:**

This paper revisits regularized policy optimization for two-player zero-sum games and proposes KLENT, a model-free reinforcement learning algorithm that combines reverse KL divergence with entropy regularization to achieve stable and efficient learning. The reviewers generally agree that the paper is technically solid, with strong theoretical analysis (including convergence guarantees) and convincing empirical results across multiple board game environments. In particular, the method demonstrates improved sample efficiency and competitive or superior performance compared to both classical search-based methods and existing regularized policy optimization approaches.

Several reviewers initially raised concerns about the level of novelty, the clarity of comparisons with prior methods, and the scope of empirical validation (e.g., restriction to board games or discrete domains). However, the authors provided thorough rebuttals, including additional experiments (e.g., comparisons with soft Q-learning and forward-KL variants) and clarifications on architectural choices and theoretical contributions, which addressed most of these concerns. Notably, multiple reviewers updated their scores upward after the rebuttal phase, acknowledging that their questions had been satisfactorily resolved.

While some limitations remain—such as moderate empirical gains in certain settings and questions about generalization beyond the studied domains—the overall contribution is clear and well-executed. The combination of theoretical grounding and practical performance makes this work a meaningful addition to the literature on reinforcement learning in games. Therefore, I recommend acceptance.